# Safety and tolerability of natural and synthetic cannabinoids in adults aged over 50 years: A systematic review and meta-analysis

**Latha Velayudhan[1], Katie McGoohan[1], Sagnik Bhattacharyya** [ID][2]*

**1** Department of Old age Psychiatry, Division of Academic Psychiatry, Institute of Psychiatry, Psychology and Neuroscience, King's College London, London, United Kingdom, **2** Department of Psychosis Studies, Division of Academic Psychiatry, Institute of Psychiatry, Psychology and Neuroscience, King's College London, London, United Kingdom

* Sagnik.2.bhattacharyya@kcl.ac.uk

**Data Availability Statement:** All relevant data are within the manuscript and its Supporting Information files.

## Abstract

### Background

Cannabinoid-based medicines (CBMs) are being used widely in the elderly. However, their safety and tolerability in older adults remains unclear. We aimed to conduct a systematic review and meta-analysis of safety and tolerability of CBMs in adults of age ≥50 years.

### Methods and findings

A systematic search was performed using MEDLINE, PubMed, EMBASE, CINAHL PsychInfo, Cochrane Library, and ClinicalTrials.gov (1 January 1990 to 3 October 2020). Randomised clinical trials (RCTs) of CBMs in those with mean age of ≥50 years for all indications, evaluating the safety/tolerability of CBMs where adverse events have been quantified, were included. Study quality was assessed using the GRADE (Grading of Recommendations Assessment, Development, and Evaluation) criteria and Preferred Reporting Items for Systematic Reviews and Meta-analyses (PRISMA) guidelines were followed. Two reviewers conducted all review stages independently. Where possible, data were pooled using random-effects meta-analysis. Effect sizes were calculated as incident rate ratio (IRR) for outcome data such as adverse events (AEs), serious AEs (SAEs), and death and risk ratio (RR) for withdrawal from study and reported separately for studies using tetrahydrocannabinol (THC), THC:cannabidiol (CBD) combination, and CBD. A total of 46 RCTs were identified as suitable for inclusion of which 31 (67%) were conducted in the United Kingdom and Europe. There were 6,216 patients (mean age 58.6 ± 7.5 years; 51% male) included in the analysis, with 3,469 receiving CBMs. Compared with controls, delta-9-tetrahydrocannabinol (THC)-containing CBMs significantly increased the incidence of all-cause and treatment-related AEs: THC alone (IRR: 1.42 [95% CI, 1.12 to 1.78]) and (IRR: 1.60 [95% CI, 1.26 to 2.04]); THC:CBD combination (IRR: 1.58 [95% CI,1.26 to 1.98]) and (IRR: 1.70 [95% CI,1.24 to 2.33]), respectively. IRRs of SAEs and deaths were not significantly greater under CBMs containing THC

**Funding:** SB is supported by grants from the National Institute of Health Research (NIHR) Efficacy and Mechanism Evaluation scheme (UK) (grant number 16/126/53) and SB and LV are in receipt of funding from Parkinson's UK (grant number G-1901). The authors acknowledge support from the National Institute for Health Research (NIHR) Biomedical Research Centre at South London and Maudsley NHS Foundation Trust and King's College London. The funders had no role in study design, data collection and analysis, decision to publish, or preparation of the manuscript.

**Competing interests:** The authors have declared that no competing interests exist.

**Abbreviations:** AEs, adverse event; CBD, cannabidiol; CBM, cannabinoid-based medicine; GRADE, Grading of Recommendations Assessment, Development, and Evaluation; IRR, incident rate ratio; MedDRA, Medical Dictionary for Regulatory Activities; PRISMA, Preferred Reporting Items for Systematic Reviews and Meta-analyses; RCT, randomised clinical trial; RR, risk ratio; SAE, serious adverse event; SOC, system organ classes; THC, delta-9-tetrahydrocannabinol.

with or without CBD. THC:CBD combination (RR: 1.40 [95% CI, 1.08 to 1.80]) but not THC alone (RR: 1.18 [95% CI, 0.89 to 1.57]) significantly increased risk of AE-related withdrawals. CBD alone did not increase the incidence of all-cause AEs (IRR: 1.02 [95% CI, 0.90 to 1.16]) or other outcomes as per qualitative synthesis. AE-related withdrawals were significantly associated with THC dose in THC only [QM (df = 1) = 4.696, $p$ = 0.03] and THC:CBD combination treatment ([QM (df = 1) = 4.554, $p$ = 0.033]. THC-containing CBMs significantly increased incidence of dry mouth, dizziness/light-headedness, and somnolence/drowsiness. Study limitations include inability to fully exclude data from those <50 years of age in our primary analyses as well as limitations related to weaknesses in the included trials particularly incomplete reporting of outcomes and heterogeneity in included studies.

## Conclusions

This pooled analysis, using data from RCTs with mean participant age ≥50 years, suggests that although THC-containing CBMs are associated with side effects, CBMs in general are safe and acceptable in older adults. However, THC:CBD combinations may be less acceptable in the dose ranges used and their tolerability may be different in adults over 65 or 75 years of age.

## Author summary

### Why was this study done?

- Use of cannabinoid-based medicines (CBMs) has been growing steadily in recent years, including in the elderly. However, their safety and tolerability in older adults remains unclear.

- With increasing interest in the use of CBMs in older people and growing unlicensed use, there is a particular need to examine their safety and tolerability in older adults.

- We analysed data on safety and tolerability from previously published double-blind, randomised controlled trials (RCT) using delta-9-tetrahydorcannabinol (THC) and cannabidiol (CBD), the common constituents of most CBMs, alone or in combination, to examine their effect on older adults.

### What did the researchers do and find?

- We pooled data from 46 published RCTs (with information from 6,216 patients; with mean participant age ≥50 years) on adverse events, serious adverse events or death, and withdrawal from study. We also examined the relationship between the dose of THC used in THC-containing CBMs and the incidence of adverse consequences in older adults.

- Our results suggest that compared with the control condition, treatment with THC-containing CBMs was associated on average with significantly greater incidence of all-cause and treatment-related adverse events.

- There was no significant increase in the incidence of serious adverse events or death with any CBMs. The risk of withdrawal from study was increased only in those receiving THC:CBD combination treatment, and this was related to THC dose.

### What do these findings mean?

- These findings suggest that CBMs in general are safe and acceptable in older adults.

- Our findings that THC-containing CBMs are associated with side effects and that THC: CBD combinations may be less acceptable at the dose ranges typically used in RCTs is critical to prescribing in older people.

## Introduction

The cannabis plant (*Cannabis sativa* L.) has been used worldwide both for recreational and medicinal purposes for thousands of years. With a fast-growing aging population, its medicinal use has also caught up and is growing in the elderly [1–3].

Among the cannabinoids found in the cannabis plant, delta-9-tetrahydorcannabinol (THC) and cannabidiol (CBD) are the most well characterised and often considered for medicinal purposes. THC can cause intoxication [4,5] and has antiemetic, analgesic, and potentially neuroprotective and anti-inflammatory effects. On the other hand, CBD is nonintoxicating [5,6] with antiepileptic and potentially also anti-inflammatory, neuroprotective, antioxidant, and antipsychotic effects [7–9]. While several trials have used these cannabinoids for a wide range of diseases and indications, a majority of these have investigated younger people [10,11]. However, age-related pharmacodynamic and pharmacokinetic changes as well as higher prevalence of comorbidities and polypharmacy in the elderly mean that they may have a different profile of safety and tolerability to cannabinoids [12,13] compared to younger people, as is well known with other groups of medications especially those used for disorders of the central nervous system [14]. Both THC and CBD, the common constituents of most cannabinoid-based medicines in current use have prominent effects on brain function and cognition [15]. Therefore, evidence of safety and tolerability of cannabinoid-based medicines (CBM) established in studies in younger adults cannot be directly extrapolated to the older adults. Although a number of recent reviews and meta-analyses [12,16,17] have summarized the safety and tolerability profile of CBMs, they have all pooled data from studies investigating across the age spectrum, making it difficult to draw age-specific inferences. With increasing interest in their use in disorders typically affecting older people [18–20] and growing unlicensed use [21], there is a particular need to investigate the safety and tolerability of CBMs in older people. This is also relevant, as there is a widely held view that many of the naturally derived cannabinoids are generally safe as they have been around and used for a long time.

Here, we have addressed this by investigating the safety and tolerability of CBMs in people over 50 years of age through systematically reviewing all double-blind, randomised controlled

trials (RCTs) using CBMs that focused on people with mean age of 50 years and over to conduct a meta-analysis. As there is a larger evidence base of studies with mean age of participants ≥50 years than the more limited set of studies that have exclusively focused on people over 50 years and even less on people over 65 or 75 years, we have focused on studies with mean age of participants ≥50 years and complemented these results with additional analyses restricted to studies that have exclusively focused on people over 50 years and even less on people over 65. Existing meta-analytic investigations [16,17] have generally considered all CBMs together, irrespective of whether they included THC, CBD, or THC:CBD in combination. However, THC can cause intoxication and may induce anxiety and transient psychotomimetic effects [5], especially at higher doses and in vulnerable individuals, while CBD does not cause intoxication when directly compared in the same individuals [5] and may potentially ameliorate anxiety and psychosis [9,22–24]. Further, there is growing evidence that THC and CBD may have opposing acute effects on autonomic arousal and brain [15] and cardiovascular function [25,26], and CBD may mitigate some of the harmful effects of THC on cognition and behaviour [15,27,28], consistent with their opposing effects on some of their molecular targets [4]. This suggests that THC and CBD may have distinct tolerability profiles, with the possibility that certain side effects may be noticeable in those taking formulations containing only THC but not in those taking formulations containing only CBD while adverse effects may even be mitigated in those taking THC and CBD in combination. This underscores the importance of examining their safety and tolerability separately. Therefore, we have addressed this issue by separately investigating the effects of THC, CBD, or THC:CBD in combination.

We hypothesized that compared to control treatments, all 3 categories of CBMs will be associated with: (i) a greater incidence of adverse events (AEs); (ii) no greater incidence of serious adverse events (SAEs) or death; and (iii) no greater risk of withdrawal from study. Further, we hypothesized a direct relationship between the dose of THC used in THC-containing CBMs and the incidence of adverse consequences in older adults.

## Methods

### Data sources and searches

The review was undertaken according to the Preferred Reporting Items for Systematic Reviews and Meta-analyses (PRISMA) reporting guidelines [29] (see S1 PRISMA checklist). The study protocol was preregistered with the International Prospective Register of Systematic Reviews (PROSPERO) (CRD42019148869). Ethics approval was not required for this systematic review and meta-analysis.

A detailed description of the bibliographic search strategy is presented in Methods in S1 Text. We identified studies published from 1 January 1990 up to 31 October 2020, from several electronic databases. Studies were independently assessed by 2 researchers and disagreements resolved through consensus or discussions with a third researcher.

### Study selection

Studies were included if (1) published from 1990 onwards; (2) included older adults (defined as mean age ≥50 years) or reported a distinct subgroup of older adults and provided separate results for this subgroup; and (3) provided data on the safety and tolerability of medical cannabinoids administered by any route, at any dose, for any duration and for any indication. Studies were excluded if they (1) included exclusively younger participants (mean age <50 years); (2) studied effects of cannabinoids for recreational purposes or failed to provide the dosage of cannabinoids; and (3) were not reported in English language. Here, we focus on results from RCTs.

## Data extraction and quality assessment

All relevant available data for examination of the safety and tolerability of different CBMs (THC:CBD combination or THC or CBD alone) were collected from eligible studies, complemented with information from ClinicalTrials.gov and author responses.

Data were extracted for study design, participant characteristics, indication, dosage and duration of intervention, all-cause and treatment-related AEs and SAEs, and AE-related withdrawals and deaths.

AEs and SAEs were coded according to the Medical Dictionary for Regulatory Activities (MedDRA) "system organ classes" (SOC). Data were also extracted for the top 5 (as reported by each study) AEs for each SOC, where available. Data extraction and coding was verified by a medically qualified researcher and discrepancies resolved following discussions with senior researcher. The disease conditions investigated were classified into broader subgroups for analysis purpose.

Overall, quality of evidence was assessed using recommended criteria [30] and summarised to reflect confidence in estimates [31].

## Data synthesis and analysis

Total exposure to active intervention in person-years was estimated by first calculating this for each individual study by multiplying the number of participants in the active intervention arm with the duration of treatment for that arm for each study and then adding up these study-specific values for all studies under each broad category (THC, THC:CBD, or CBD) of intervention investigated here. Mean exposure in person-years for each category (THC, THC:CBD, or CBD) of intervention was estimated by calculating the arithmetic mean from study-specific estimates obtained as above for each intervention category. Pooled mean ages of participants for each group of studies and treatment arms were estimated by calculating the arithmetic mean of study specific mean age as reported by individual studies for each intervention arm. Other pooled estimates (median and interquartile range) for summary study characteristics, such as duration of study in weeks, participants analysed or included, or duration of treatment for each treatment arm, were calculated by estimating them from the total number (e.g., for variables such as number of participants included or analysed; duration of study or treatment) or the mean estimate (e.g., for variables such as mean age) as reported in individual studies for each set subgroups of RCTs.

Pooled effect-sizes were estimated if there were 2 or more RCTs within each group or subgroup under the random-effects model using the restricted maximum-likelihood estimator because of anticipated heterogeneity. For each broad category of intervention, analyses combined both parallel-arm and crossover RCTs, with the latter treated as parallel-arm design [32] for pooled analyses. But we also report results by RCT design for each intervention. We estimated incident rate ratio (IRR) for outcome data such as AEs, SAEs, and death and risk ratio (RR) for withdrawal from study. Studies with more than 1 active treatment arm were treated as independent studies. In studies with more than 1 active treatment arm compared against a single control group, we also report meta-analysis of dependent effect-sizes with robust variance estimation [33–35]. For the purpose of reporting throughout the manuscript, results are reported for analyses treating all studies as independent, while corresponding dependent meta-analyses are reported in Results in S1 Text and signposted in the main text as appropriate. We investigated heterogeneity using forest plots and the $I^2$ statistic and publication bias using Egger regression test [36] and the "Trim and fill" method [37]. For the analysis of AEs, data for all conditions were combined. We also examined whether estimates varied according to treatment, design, clinical condition, and dose of study drug using meta-regression.

Our primary analysis includes the results of all studies where the mean age of study participants was ≥50 years. As many participants in these included studies were <50 years of age, we also carried out separate analyses restricted to studies where all participants were ≥50 years of age and also where all participants were ≥65 years of age. These analyses were carried out where there were at least 2 studies with analysable data. Statistical analyses were performed using the metafor package in R (version 3.6.3) [38]. For meta-analysis of dependent effect-sizes with robust variance estimation, we also used the *clubSandwich* package (https://github.com/jepusto/clubSandwich) along with *metafor* in R.

## Results

### Data selection

Fig 1 (PRISMA flow chart) summarizes the study selection procedure. Main characteristics and outcome measures of each study are included in Tables 1–3; additional details regarding

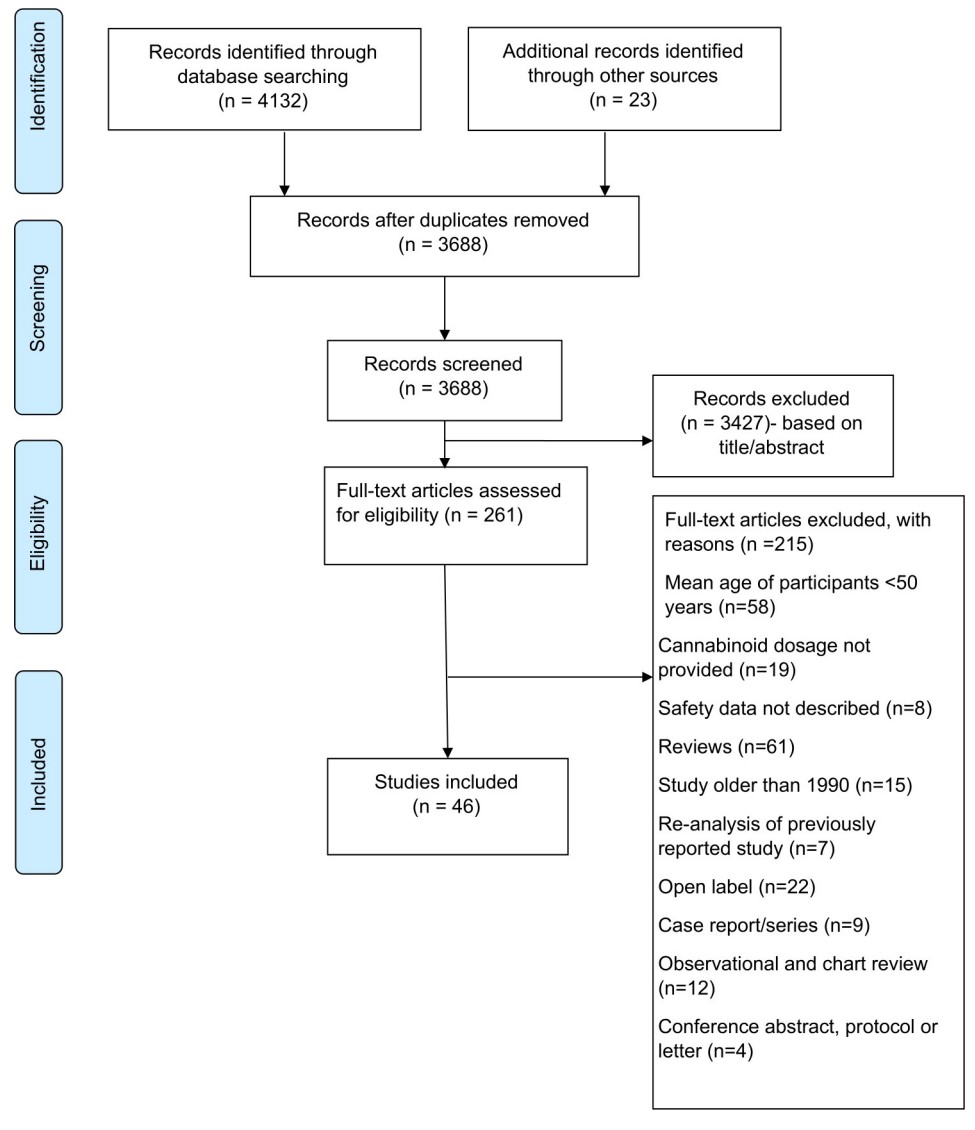

**Fig 1. Study disposition.**

Table 1. Characteristics of included randomised controlled trials of THC in older adults (N = 30).

| Study ID (country) | Study Design (RCT) | THC: Sample included/ analysed N Mean age (SD), Male % | Comparator: Sample included/ analysed N Mean age (SD), Male % | Age cut-off for enrolment | Indication | THC classification | Comparator | THC treatment duration, weeks | Calculated daily average THC dose | Overall GRADE rating for study |
|---|---|---|---|---|---|---|---|---|---|---|
| ***Ahmed et al. 2014 (the Netherlands) | Crossover | 12/11 72.00 (5), 50 | 12/11 72.00 (5), 50 | ≥65 years | Healthy older participants | Namisol | Placebo | .4 | 6.5 mg | Moderate |
| Ahmed et al. 2015 (the Netherlands) | Crossover | 10/10 77.30 (5.6), 70 | 10/10 77.30 (5.6), 70 | ≥18 years | Dementia | Namisol | Placebo | 2.6 | 3 mg | Moderate |
| Brisbois et al. 2011 (Canada) | Parallel-arm | 24/11 67.00 (10.9), 64 | 22/10 65.50 (8), 50 | Adult | Cancer patients with chemosensory alterations | Dronabinol | Placebo | 2.6 | 7.5 mg | Low |
| Carley et al. 2018 (USA)‡ | Parallel-arm | 21/21 52.70 (7.7), 76 | 25/25 58.80 (6.1), 72 | 21–65 years | Obstructive sleep apnoea | Dronabinol | Placebo | 6.0 | 2.5 mg | Low |
| Carley et al. 2018 (USA)‡ | Parallel-arm | 27/27 54.70 (7), 67 | 25/25 58.80 (6.1), 72 | 21–65 years | Obstructive sleep apnoea | Dronabinol | Placebo | 6.0 | 10 mg | Low |
| Curtis et al. 2009 (UK) | Crossover | 44/37 52.00 (9.5), 50 | 44/37 52.00 (9.5), 50 | ≥18 years | Huntington disease | Nabilone | Placebo | 5.0 | 2 mg | Low |
| De Vries et al. 2016 (the Netherlands) | Crossover | 25/24 52.00 (NR), 62 | 25/24 52.00 (NR), 62 | >18 years | Chronic pancreatitis | Namisol | Diazepam | .1 | 8 mg | Moderate |
| **Herrmann et al. 2019 (Canada) | Crossover | 39/38 87.00 (10), 77 | 39/38 87.00 (10), 77 | ≥55 years | Alzheimer disease | Nabilone | Placebo | 6.0 | 1.6 mg | Moderate |
| Jatoi et al. 2002 (USA) | Parallel-arm | 152/152 67.00 (10), 66 | 159/159 65.00 (11), 65 | ≥18 years | Cancer-related anorexia | Dronabinol | Megestrol acetate | 8.1 | 5 mg | Low |
| Johnson et al. 2010 (UK)§ | Parallel-arm | 58/58 61.30 (12.5), 52 | 59/59 60.10 (12.3), 54 | NR | Patients with cancer-related pain | THC extract spray | Placebo | 2.0 | 23 mg | Moderate |
| Lane et al. 1991 (USA) | Parallel-arm | 21/21 47.0 (20–68)*,† 48 | 21/21 49.0 (22–64)*,† 48 | 18–69 years | Chemotherapy-induced nausea and vomiting | Dronabinol | Prochlorperazine | .9 | 40 mg | Low |
| Meiri et al. 2007 (USA) | Parallel-arm | 17/17 61.60 (14.2), 53 | 14/14 57.20 (8.6), 38 | ≥18 years | Chemotherapy-induced nausea and vomiting | Dronabinol | Placebo | .7 | 20 mg | Low |
| Peball et al. 2020 (Austria) | Parallel-arm | 19/19 65.4 (7.94), 53 | 19/19 64.0 (8.04), 74 | ≥30 years | Parkinson disease | Nabilone | Placebo | 4.0 | 0.75 mg | Moderate |
| Sieradzan et al. 2001 (UK) | Crossover | 9/9 59.00 (NR), 44 | 9/9 59.00 (NR), 44 | NR | Parkinson disease | Nabilone | Placebo | .1 | 2 mg | Very low |
| Strasser et al. 2006 (Germany)§ | Parallel-arm | 100/100 60.00 (12), 54 | 48/48 62.00 (10), 52 | Adult | Cancer-related anorexia | THC | Placebo | 6.0 | 5 mg | Low |
| Svendsen et al. 2004 (Denmark) | Crossover | 24/24 50.0 (NR), 42 | 24/24 50.0 (NR), 42 | 18–55 years | Multiple sclerosis | Dronabinol | Placebo | 3.0 | 10 mg | Moderate |
| Tomida et al. 2006 (UK)§ | Crossover | 6/6 55.30 (5), 100 | 6/6 55.30 (5), 100 | NR | Intraocular pressure | THC extract spray | Placebo | .1 | 5 mg | Low |
| Toth et al. 2012 (Canada) | Parallel-arm | 13/13 60.80 (15.3), 38 | 13/13 61.60 (14.6), 69 | 18–80 | Diabetic peripheral neuropathic pain | Nabilone | Placebo | 5.0 | 4 mg | Low |

(Continued)

**Table 1.** (Continued)

| Study ID (country) | Study Design (RCT) | THC: Sample included/ analysed N Mean age (SD), Male % | Comparator: Sample included/ analysed N Mean age (SD), Male % | Age cut-off for enrolment | Indication | THC classification | Comparator | THC treatment duration, weeks | Calculated daily average THC dose | Overall GRADE rating for study |
|---|---|---|---|---|---|---|---|---|---|---|
| **Van Amerongen et al. 2017, 2 (the Netherlands)‖** | Crossover | 24/24 54.30 (8.9), 33 | 24/24 54.30 (8.9), 33 | ≥18 years | Multiple sclerosis | THC | Placebo | .1 | 16 mg | Moderate |
| **Van Amerongen et al. 2017, 1 (the Netherlands)‖** | Parallel-arm | 12/12 57.30 (9), 33 | 12/12 51.40 (8), 33 | ≥18 years | Multiple sclerosis | THC | Placebo | 4.0 | 28.5 mg | Moderate |
| **Van den Elsen et al. 2015, 1 (the Netherlands)** | Parallel-arm | 24/24 79.00 (8), 46 | 26/26 78.00 (7), 54 | ≥40 years | Dementia | Namisol | Placebo | 3.0 | 4.5 mg | Moderate |
| **Van den Elsen et al. 2015, 2 (the Netherlands)** | Crossover | 22/22 76.40 (5.3), 68 | 22/22 76.40 (5.3), 68 | ≥18 years | Dementia | Namisol | Placebo | 2.6 | 3 mg | Moderate |
| ***Volicer et al. 1997 (USA)** | Crossover | 15/12 72.70 (4.9), 92 | 15/12 72.70 (4.9), 92 | NR | Alzheimer disease | Dronabinol | Placebo | 6.0 | 5 mg | Very low |
| ***Walther et al. 2011 (Switzerland)** | Crossover | 2/2 78.00 (NR), 100 | 2/2 78.00 (NR), 100 | NR | Alzheimer disease | Dronabinol | Placebo | 2.0 | 2.5 mg | Very low |
| **Ware et al. 2010 (Canada)** | Crossover | 32/32 50.00 (11.2), 16 | 32/32 50.00 (11.2), 16 | ≥18 years | Fibromyalgia | Nabilone | Amitriptyline | 2.0 | 1 mg | Moderate |
| **Weber et al. 2010 (Switzerland)** | Crossover | 27/22 57.00 (12), 74 | 27/22 57.00 (12), 74 | Adult | Amyotrophic lateral sclerosis patients with cramps | Dronabinol | Placebo | 2.0 | 10 mg | Moderate |
| **Zadikoff et al. 2011 (Canada)** | Crossover | 9/9 60.00 (7), 0 | 9/9 60.00 (7), 0 | 18–75 | Cervical dystonia | Dronabinol | Placebo | 3.0 | 15 mg | Low |
| **Zajicek et al. 2003 (UK)§** | Parallel-arm | 216/206 50.00 (8.2), 31 | 222/213 51.00 (7.6), 37 | 18–64 | Multiple sclerosis | Dronabinol | Placebo | 14.0 | 25mg | Moderate |
| **Zajicek et al. 2005 (UK)§** | Parallel-arm | 125/125 50.00 (8.2), 31 | 120/120 51.00 (7.6), 37 | 18–64 | Multiple sclerosis | Dronabinol | Placebo | 52.0 | 25 mg | Moderate |
| **Zajicek et al. 2013 (UK)** | Parallel-arm | 332/329 52.30 (7.6), 40 | 166/164 52.00 (8.2), 41 | 18–65 | Multiple sclerosis | Dronabinol | Placebo | 160.0 | 28 mg | Moderate |

***Studies recruited participants ≥65 years.

‡Article included more than 1 dose level.

§Article included more than 1 cannabinoid intervention.

*Median age (range).

†Included as median age for whole study population was ≥50.

‖Article included the results of multiple trials.

GRADE, Grading of Recommendations Assessment, Development, and Evaluation; NR, not recorded; RCT, randomised clinical trial; THC, delta-9-tetrahydrocannabinol.

studies are presented in Results in S1 Text and Tables A and B in S1 Text. A total of 60 comparisons of CBM and control intervention using RCT design (hereafter called RCTs) ($n$ = 6,216 participants; 1933.47 person-years of cannabinoid exposure) from 46 published articles

**Table 2. Characteristics of included randomised controlled trials of THC:CBD combination in older adults (*N* = 26).**

| Study ID (country) | Study Design | CBD/THC: Sample included/ analysed N Mean age (SD), Male % | Comparator: Sample included/ analysed N Mean age (SD), Male % | Age cut-off for enrolment | Indication | CBD/THC classification | Comparator | CBD/THC treatment duration, weeks | Calculated daily average CBD/THC dose | GRADE rating |
|---|---|---|---|---|---|---|---|---|---|---|
| **Blake et al. 2006 (UK)** | Parallel-arm | 31/31 60.9 (10.6), 26 | 27/27 64.9 (8.5), 15 | NR | Rheumatoid arthritis | THC:CBD spray | Placebo | 5.0 | 14.6 mg THC: 13.5 mg CBD | Low |
| **\*\*Carroll et al. 2004 (UK)** | Crossover | 19/17 67.0 (NR) 63 | 19/17 67.0 (NR) 63 | 18–78 years | Levodopa-induced dyskinesia in Parkinson disease | Cannabis extract | Placebo | 4.0 | 10.2 mg THC: 5.1 mg CBD | Moderate |
| **Duran et al. 2010 (Spain)** | Parallel-arm | 7/7 50 (41–70)\* 0 | 9/9 50 (34–76)\* 11 | >18 years | Chemotherapy-induced nausea and vomiting | THC:CBD spray | Placebo | .6 | 13 mg THC: 12 mg CBD | Moderate |
| **Fallon et al. 2017, 1 (Multicentre)‖** | Parallel-arm (withdrawal study) | 103/103 61.4 (10.9), 61 | 103103 61.6 (11.8), 53 | ≥18 years | Advanced cancer patients with pain | THC:CBD spray | Placebo | 5.0 | 17.6 mg THC: 16.3 mg CBD | Moderate |
| **Fallon et al. 2017, 2 (Multicentre)‖** | Parallel-arm | 200/199 60.0 (11), 53 | 199/198 59.6 (11), 49 | ≥18 years | Advanced cancer patients with pain | THC:CBD spray | Placebo | 5.0 | 17 mg THC: 15.8 mg CBD | Moderate |
| **Jadoon et al. 2016, 1 (UK)‡,§** | Parallel-arm | 11/11 59.0 (8.8), 55 | 14/14 59.0 (7.7), 50 | ≥18 years | Type 2 diabetes | CBD/THCV | Placebo | 13.0 | 10 mg THC: 10 mg CBD | Moderate |
| **Jadoon et al. 2016, 2 (UK)‡,§** | Parallel-arm | 12/12 58.0 (8.1), 75 | 14/14 59.0 (7.7), 50 | ≥18 years | Type 2 diabetes | CBD/THCV | Placebo | 13.0 | 10 mg THC: 200 mg CBD | Moderate |
| **Johnson et al. 2010, (UK)§** | Parallel-arm | 60/60 59.4 (12.1), 55 | 59/59 60.1 (12.3), 54 | NR | Patients with cancer-related pain | THC:CBD spray | Placebo | 2.0 | 25 mg THC: 23 mg CBD | Moderate |
| **Litchman et al. 2018, (Multicentre)** | Parallel-arm | 199/199 59.2 (12), 56 | 198/198 60.7 (11.1), 52 | ≥18 years | Advanced cancer patients with pain | THC:CBD spray | Placebo | 5.0 | 17.3 mg THC: 16 mg CBD | Moderate |
| **Lynch et al. 2014 (USA)** | Crossover | 18/16 56.0 (10.8), 17 | 18/16 56.0 (10.8), 17 | NR | Chemotherapy-induced neuropathic pain | THC:CBD spray | Placebo | 6.0 | 21.6 mg THC: 20 mg CBD | Low |
| **Markova et al. 2019 (Czech Republic)** | Parallel-arm | 53/53 51.3 (10.2) 30 | 53/53 51.3 (10.2) 30 | ≥18 years | Multiple sclerosis | THC:CBD spray | Placebo | 12.0 | 19.7 mg THC: 18.3 mg CBD | Low |
| **Notcutt et al. 2012 (UK)** | Parallel-arm (withdrawal study) | 18/18 59.7 (9) 50 | 18/18 54.4 (10.4) 33 | NR | Multiple sclerosis | THC:CBD spray | Placebo | 4.0 | 20.8 mg THC: 19.3 mg CBD | Very low |
| **Nurmikko et al. 2007 (UK)** | Parallel-arm | 63/63 52.4 (15.8), 44 | 62/62 54.3 (15.2), 37 | ≥18 years | Neuropathic pain | THC:CBD spray | Placebo | 5.0 | THC 29.7 mg: CBD 27.5 mg | High |
| **\*\*\*Pickering et al. 2011, 1 (UK)¶** | Crossover | 5/4 67.0 (NR), 50 | 5/4 67.0 (NR), 50 | 40–74 years | COPD | THC:CBD spray | Placebo | .1 | 4.7 mg THC: 4.4 mg CBD | Low |
| **\*\*Pickering et al. 2011, 2 (UK)¶** | Crossover | 6/5 58.0 (NR), 80 | 6/5 58.0 (NR), 80 | 40–75 years | Healthy controls | THC:CBD spray | Placebo | .1 | 10.3 mg THC: 9.5 mg CBD | Low |
| **Portenoy et al. 2012, 1 (Multicentre)‡** | Parallel-arm | 91/91 59.0 (12.3), 49 | 91/91 56.0 (12.2), 48 | NR | Cancer patients with chronic pain | THC:CBD spray | Placebo | 5.0 | 10.8 mg THC: 10 mg CBD | Moderate |

(*Continued*)

**Table 2.** (Continued)

| Study ID (country) | Study Design | CBD/THC: Sample included/ analysed N Mean age (SD), Male % | Comparator: Sample included/ analysed N Mean age (SD), Male % | Age cut-off for enrolment | Indication | CBD/THC classification | Comparator | CBD/THC treatment duration, weeks | Calculated daily average CBD/THC dose | GRADE rating |
|---|---|---|---|---|---|---|---|---|---|---|
| **Portenoy et al. 2012, 2 (Multicentre)**‡ | Parallel-arm | 88/87 59.0 (13.1), 56 | 91/91 56.0 (12.2), 48 | NR | Cancer patients with chronic pain | THC:CBD spray | Placebo | 5.0 | 27 mg THC: 25 mg CBD | Moderate |
| **Portenoy et al. 2012, 3 (Multicentre)**‡ | Parallel-arm | 90/90 58.0 (11.2), 53 | 91/91 56.0 (12.2), 48 | NR | Cancer patients with chronic pain | THC:CBD spray | Placebo | 5.0 | 43.2 mg THC: 40 mg CBD | Moderate |
| **Riva et al. 2019 (Italy)** | Parallel-arm | 30/29 58.4 (10.6) 62 | 30/30 57.2 (13.8) 53 | 18–80 years | Motor neurone disease | THC:CBD spray | Placebo | 6.0 | 21.6 mg THC: 20.0 mg CBD | High |
| **Serpell et al. 2014 (UK)** | Parallel-arm | 128/128 57.6 (14.4), 34 | 118/118 57.0 (14.1), 45 | ≥18 years | Neuropathic pain | THC:CBD spray | Placebo | 14.0 | 24 mg THC: 22 mg CBD | Moderate |
| **Strasser et al. 2006 (Germany)**§ | Parallel-arm | 95/95 61.0 (12), 56 | 48/48 62.0 (10), 52 | Adult | Cancer-related anorexia | Cannabis extract | Placebo | 6.0 | 5 mg THC: 2 mg CBD | Moderate |
| **Vaney et al. 2004 (Switzerland)** | Crossover | 57/50 55.0 (10), 49 | 57/50 55.0 (10), 49 | Adult | Multiple sclerosis | Cannabis extract | Placebo | 2.0 | 27.5 mg THC: 9.9 mg CBD | Low |
| **Wade et al. 2004 (UK)** | Parallel-arm | 80/80 51.0 (9.4), 41 | 80/80 50.0 (9.3), 35 | NR | Multiple sclerosis | THC:CBD spray | Placebo | 6.0 | 40.5 mg THC: 37.5 mg CBD | Moderate |
| **Zajicek et al. 2003 (UK)**§ | Parallel-arm | 219/211 51.0 (7.6), 36 | 222/213 51.0 (7.6), 37 | 18–64 years | Multiple sclerosis | Cannabis extract | Placebo | 14.0 | 25 mg THC: 12.5 mg CBD | Moderate |
| **Zajicek et al. 2005 (UK)**§ | Parallel-arm | 138/138 51.0 (7.6), 36 | 120/120 51.0 (7.6), 37 | 18–64 years | Multiple sclerosis | Cannabis extract | Placebo | 52.0 | 25 mg THC: 12.5 mg CBD | Moderate |
| **Zajicek et al. 2012 (UK)** | Parallel-arm | 144/143 51.9 (7.7), 39 | 135/134 52.0 (7.9), 35 | 18–64 years | Multiple sclerosis | Cannabis extract | Placebo | 12.0 | 25 mg THC: 12.5 mg CBD | Moderate |

**Studies recruited participants ≥50years.

*Median age (range).

‖Article included the results of multiple trials.

‡Article included more than 1 dose level.

§Article included more than 1 cannabinoid intervention.

***Studies recruited participants ≥65 years.

ꬻArticle included multiple study groups/indications.

CBD, cannabidiol; COPD, chronic obstructive pulmonary disease; GRADE, Grading of Recommendations Assessment, Development, and Evaluation; NR, not recorded; THC, delta-9-tetrahydrocannabinol; THCV, tetrahydrocannabivarin.

were included (Fig 1). Of these, 4 RCTs recruited participants over age ≥65 years (*n* = 68; mean age, 72.4 (SD ± 4.5)), of which one was ≥75 years (Tables 1 and 2) [39–42].

The formulations used in THC studies were (numbers within brackets indicating the number of RCTs where each formulation was used): nabilone (6), dronabinol (marinol) (14), THC (3), THC extract spray (2), and Namisol (5). The combination THC:CBD trials used THC:

**Table 3. Characteristics of randomised controlled trials of CBD in older adults (N = 4).**

| Study ID (country) | Study Design | CBD: Sample included/ analysed N Mean age (SD) Male % | Comparator: Sample included/ analysed N Mean age (SD) Male % | Age cut-off for enrolment | Indication | Active treatment | Comparator | CBD treatment duration, weeks | Calculated daily average CBD dose | GRADE rating |
|---|---|---|---|---|---|---|---|---|---|---|
| **Consroe et al. 1991 USA** | Crossover | 18/15 47.8$^2$ (15.3), 53 | 18/15 47.8$^2$ (15.3), 53 | NR | Huntington disease | CBD | Placebo | 6 | 700 mg | Low |
| **Jadoon et al. 2016[§] UK** | Parallel-arm | 13/13 56.8 (9.9), 77 | 14/14 59.0 (9.4) 68 | ≥18 years | Type 2 diabetes | CBD | Placebo | 13 | 200 mg | Moderate |
| **Tomida et al. 2006[‡,§] UK** | Crossover | 6/6 55.3 (5.0), 100 | 6/6 55.3 (5.0), 100 | NR | Intraocular pressure | CBD | Placebo | 0.1 | 20 mg | Low |
| **Tomida et al. 2006[‡,§] UK** | Crossover | 6/6 55.3 (5.0), 100 | 6/6 55.3 (5.0), 100 | NR | Intraocular pressure | CBD | Placebo | 0.1 | 40 mg | Low |

§Article included more than 1 cannabinoid intervention.

‡Article included more than 1 dose level.

CBD, cannabidiol; GRADE, Grading of Recommendations Assessment, Development, and Evaluation; NR, not recorded.

CBD spray (18), cannabis extract (6), and CBD/THCV (2). The CBD studies used CBD preparations only.

The disease conditions investigated were classified into broader subgroups for analysis purpose as neurodegenerative (Alzheimer disease, Parkinson disease, Huntington disease, amyotrophic lateral sclerosis), multiple sclerosis, motor neurone disease, pain (neuropathic pain), cancer (cancer or chemotherapy-related anorexia, pain, or nausea/vomiting), and others (type 2 diabetes mellitus, chronic obstructive pulmonary disease, fibromyalgia, raised intraocular pressure, cervical dystonia, healthy, pancreatitis, obstructive sleep apnoea, and Levodopa-induced dyskinesia in Parkinson disease).

Figs 2–14 show the forest plots and summary results of the meta-analyses stratified according to study design, for all-cause and treatment-related AEs and SAEs, withdrawals, deaths, for studies using THC, THC:CBD combination, and CBD, respectively.

## THC studies

A total of 30 RCTs (15 crossover and 15 parallel-arm) from 28 articles [39–41,43–67] (see Results in S1 Text and Table A in S1 Text, for additional details), reported on 1,461 patients on active [analysed 1,417; Total person-years of THC exposure: 1252.83 person-years; Mean person-years of THC exposure (mean ± SD): 41.76 ± 184.28 person-years] and 1,251 (analysed 1,210) patients on control intervention, ranging from 50 to 87 years in mean age (males: 0% to 100%). All except 4 studies used placebo control [43,45,54,62].

Pooled IRRs for all-cause (k = 21) and treatment-related AEs (k = 9) from all RCTs were 1.42 (95% CI, 1.12 to 1.79) and 1.60 (95% CI, 1.26 to 2.04), respectively. Pooled IRRs for all-cause (k = 27) and treatment-related (k = 23) SAEs from all RCTs were 1.08 (95% CI, 0.80 to 1.46) and 1.23 (95% CI, 0.56 to 2.69), respectively. Pooled RR for AE-related withdrawals

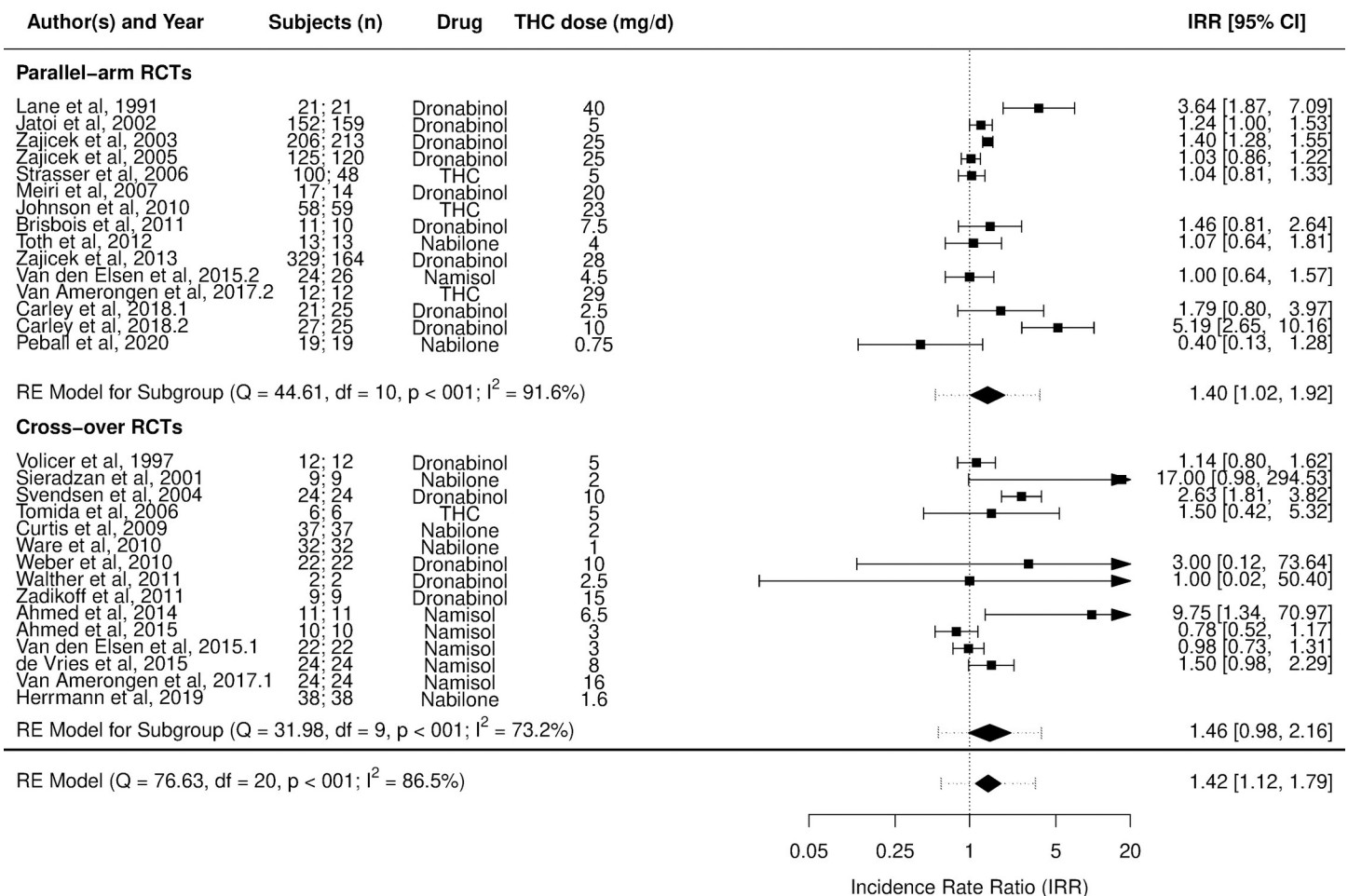

**Fig 2. Forest plot of all-cause adverse events: THC studies.** Numbers under the "Subjects (n)" column refer to analysed participants from the active and control intervention arms, respectively. IRR, incident rate ratio; RCT, randomised clinical trial; THC, delta-9-tetrahydrocannabinol.

($k$ = 27) and IRR for all deaths ($k$ = 30) from all RCTs were 1.18 (95% CI, 0.89 to 1.57) and 1.09 (95% CI, 0.75 to 1.59), respectively. Neither Egger test nor "Trim and fill" method indicated publication or other selection bias for any of the outcomes except for SAEs (Results in S1 Text and Fig A in S1 Text). For all-cause SAEs, while Egger test was nonsignificant, Trim-fill method identified 10 missing studies. The estimated effect of treatment on IRR for all-cause SAEs, which was not significant previously, became significant after inclusion of potentially missing studies identified by the Trim-fill method (1.46, 95% CI, 1.09 to 1.95, $p$ = 0.01, $k$ = 35). For treatment-related SAEs, Egger test indicated significant publication bias and Trim-fill method identified 5 missing studies, though they did not change the direction or significance of effect-size on inclusion. Where there was nonindependence of outcome data used in analyses, results of dependent meta-analyses were consistent with the results of independent meta-analyses (Results in S1 Text).

**Effect of moderators.** Meta-regression analyses indicate that there was a trend-level effect [QM (df = 4) = 9.986, $p$ = 0.084] of clinical condition on estimated effect of THC treatment on all-cause AEs, which seemed to be mainly related to a significantly lower estimated effect in RCTs investigating neurodegenerative disorder (regression coefficient = 0.905; $p$ = 0.006)

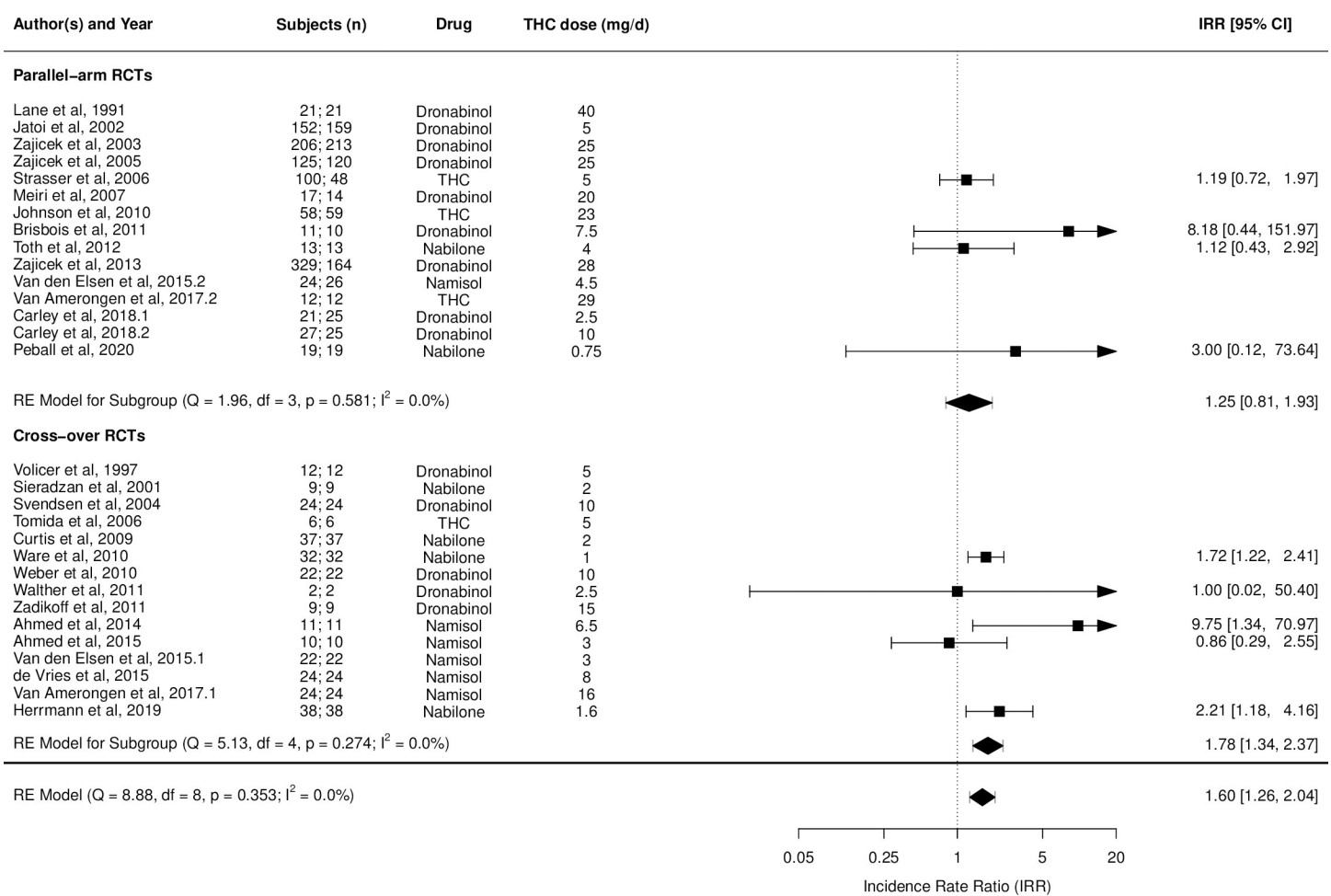

**Fig 3. Forest plot of treatment-related adverse events: THC studies.** Numbers under the "Subjects (n)" column refer to analysed participants from the active and control intervention arms, respectively. IRR, incident rate ratio; RCT, randomised clinical trial; THC, delta-9-tetrahydrocannabinol.

patients compared to other conditions. Except that, moderators such as study design or type of intervention did not significantly influence estimated effects of THC treatment on any of the outcomes assessed.

**Effect of dose.** Meta-regression analyses also indicated that there was a significant effect of daily THC dose on all-cause AEs [QM (df = 1) = 5.024, $p$ = 0.025] as well as on AE-related withdrawals [QM (df = 1) = 4.696, $p$ = 0.03] for all RCTs indicating that higher the dose of THC the higher was the risk of all-cause AEs and risk of withdrawal (regression coefficient = −0.905; $p$ = 0.006) from study in THC-treated patients compared to control treatment (Fig B in S1 Text). There was no significant association of daily THC dose with any of the other estimates (SAEs and deaths).

**Common side effects.** Pooled IRRs of the most commonly reported AEs (Table C in S1 Text) suggested significantly higher incidence rate of dry mouth, dizziness/light-headedness, mobility/balance/coordination difficulties, somnolence/drowsiness, euphoria, and male impotence in active compared to control arms.

**Analysis of studies where all participants were ≥50 years.** Restricting the meta-analysis to the 4 studies [39–41,65] that recruited participants with ≥50 years of age (total $n$ = 136; analysed $n$ = 126), results were broadly comparable to the primary analysis in the pattern of

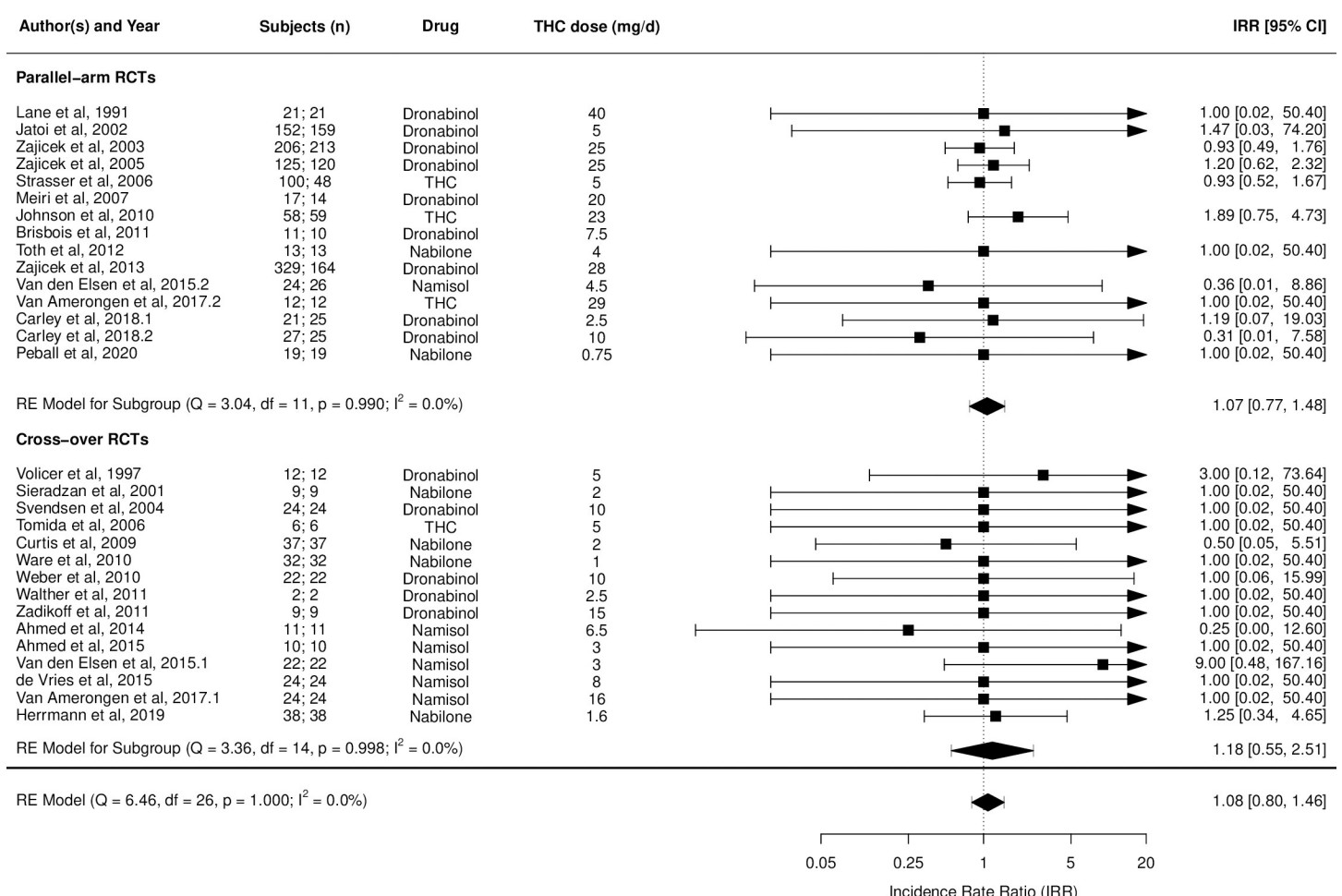

**Fig 4. Forest plot of all-cause serious adverse events: THC studies.** Numbers under the "Subjects (n)" column refer to analysed participants from the active and control intervention arms, respectively. IRR, incident rate ratio; RCT, randomised clinical trial; THC, delta-9-tetrahydrocannabinol.

findings, though effect on all-cause AEs was no longer significant. The pooled IRRs for all-cause AEs (k = 3) were 2.13 (95% CI, 0.46 to 9.97) and treatment-related AEs (k = 3) was 2.80 (95% CI, 1.09 to 7.21), respectively. Pooled IRRs for all-cause (k = 4) and treatment-related SAEs (k = 2) were 1.20 (95% CI, 0.39 to 3.65) and 0.50 (95% CI, 0.03 to 7.99), respectively. Pooled RR for AE-related withdrawals (k = 4) and IRR for all deaths (k = 4) were 1.11 (95% CI, 0.49 to 2.55) and 0.58 (95% CI, 0.11 to 3.06), respectively (Fig C–H in S1 Text).

**Analysis of studies where all participants were ≥65 years.** Restricting the meta-analysis to only the 3 studies [39–41] that recruited participants with ≥65 years of age (total n = 58; analysed n = 50), effects were also broadly comparable to the primary analysis in the pattern of findings, with the exception of effect on all-cause AEs, which was no longer significant. The pooled IRRs for all-cause AEs (k = 3) were 2.13 (95% CI, 0.46 to 9.97) and treatment-related AEs (k = 2) was 2.80 (95% CI, 1.09 to 7.21), respectively. Pooled IRRs for all-cause (k = 3) and treatment-related SAEs (k = 2) were 1.80 (95% CI, 0.13 to 8.76) and 0.50 (95% CI, 0.03 to 7.99), respectively. Pooled RR for AE-related withdrawals (k = 3) and IRR for all deaths (k = 3) were 1.00 (95% CI, 0.23 to 4.41) and 0.42 (95% CI, 0.05 to 3.42), respectively.

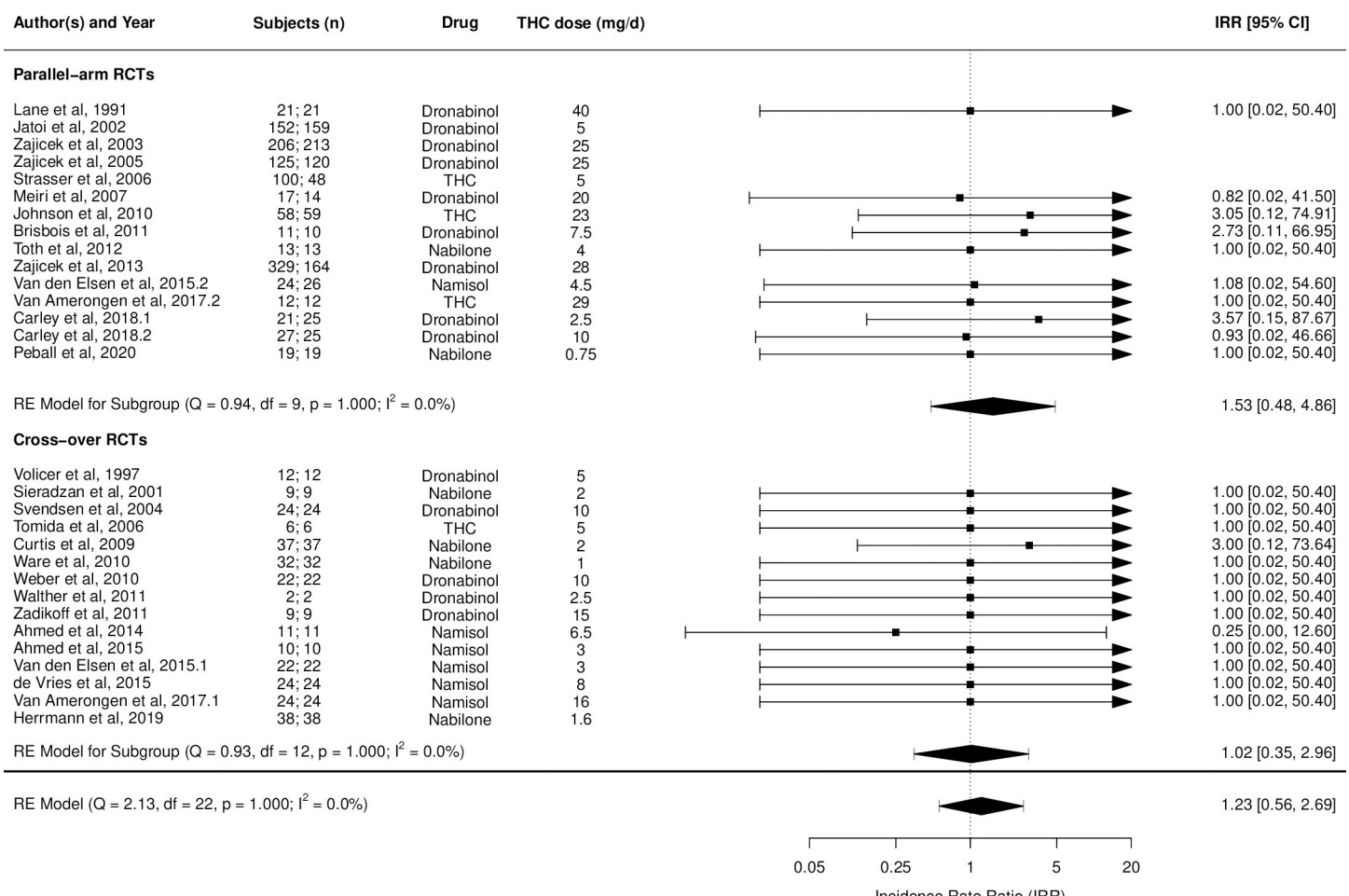

**Fig 5. Forest plot of treatment-related serious adverse events: THC studies.** Numbers under the "Subjects (n)" column refer to analysed participants from the active and control intervention arms, respectively. IRR, incident rate ratio; RCT, randomised clinical trial; THC, delta-9-tetrahydrocannabinol.

## THC:CBD combination studies

A total of 26 studies (5 crossover and 21 were parallel-arm; see Results in S1 Text and Table B in S1 Text for additional details) from 21 articles [42,46,48,49,68–84] reported on 1,965 patients [analysed 1,940; Total person-years of THC:CBD exposure: 394.29 person-years; Mean person-years of THC exposure (mean ± SD): 15.17 ± 28.20 person-years] on active and 1,887 (analysed 1,863) on placebo, ranging from 50 to 67 years in age (males: 0% to 80%). All studies used placebo as control. Two of the 26 included studies [80] investigated a combination of CBD and THCV, and we also examined the key effects after excluding these studies, which remained unchanged (please see Fig I–N in S1 Text). Results of meta-analysis for individual AEs after excluding these studies were identical to the results including all studies (as shown in Table D in S1 Text) and hence not shown.

Pooled IRRs for all-cause ($k = 16$) and treatment-related ($k = 9$) AEs from all RCTs were 1.58 (95% CI, 1.26 to 1.98) and 1.70 (95% CI, 1.24 to 2.33), respectively. Pooled IRRs for all-cause ($k = 26$) and treatment-related ($k = 21$) SAEs from all RCTs were 1.17 (95% CI, 0.99 to 1.39) and 1.19 (95% CI, 0.88 to 1.62), respectively. Pooled RR for AE-related withdrawals ($k = 26$) and IRR for all deaths ($k = 26$) from all RCTs were 1.40 (95% CI, 1.08 to 1.80) and 1.14 (95% CI, 0.89 to 1.46), respectively. Neither Egger test nor "Trim and fill" method indicated

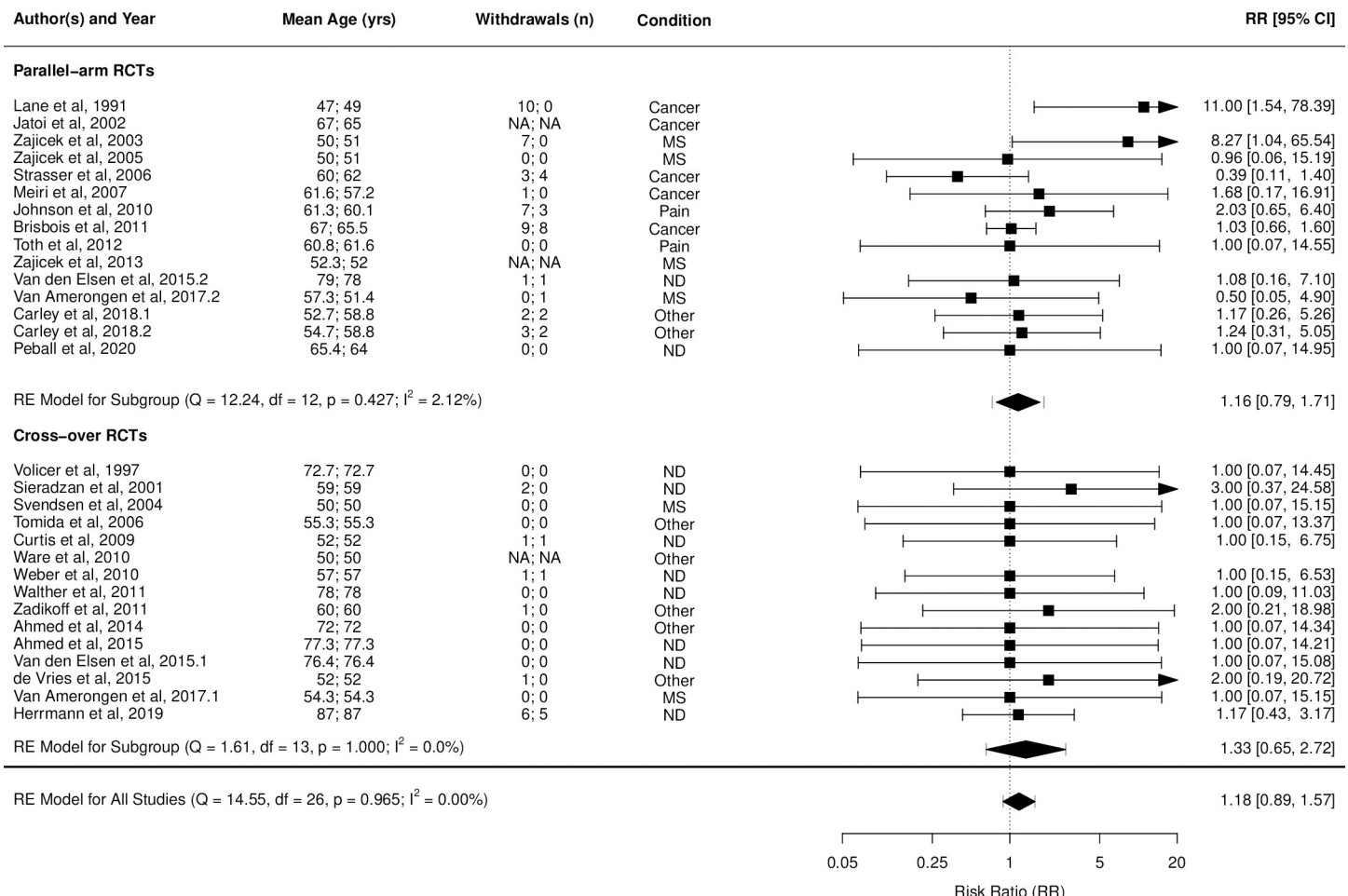

**Fig 6. Forest plot of adverse event-related withdrawals: THC studies.** Numbers under the "Mean Age (yrs)" and "Withdrawals (n)" columns refer to the values in active and control intervention arms, respectively. The conditions listed are the disease conditions subgrouped into broader categories for meta-regression analyses purposes. They are: ND (dementia, Alzheimer disease, PD, Huntington disease, amyotrophic lateral sclerosis); MS; Cancer (cancer or chemotherapy-related anorexia, pain or nausea/vomiting, chemosensory alterations); and Other (type 2 diabetes mellitus, fibromyalgia, raised intraocular pressure, cervical dystonia, healthy, pancreatitis, obstructive sleep apnoea). MS, multiple sclerosis; NA, Not Available; ND, neurodegenerative disease; PD, Parkinson disease; RCT, randomised clinical trial; RR, risk ratio; THC, delta-9-tetrahydrocannabinol.

any significant effect of publication or other selection bias for any of the outcomes except for deaths (Fig O in S1 Text). For deaths as outcome, while Egger test was not significant, Trim and fill method indicated 9 missing studies, with the estimated effect becoming significant after their inclusion (1.33, 95% CI: 1.03 to 1.71; $p = 0.027$, $k = 35$). Where there was nonindependence of outcome data used in analyses, results of dependent meta-analyses were consistent with the results of independent meta-analyses.

**Effect of moderators.** Meta-regression analysis indicated that there was a significant effect of clinical condition on effect-size for treatment-related AEs [QM (df = 3) = 15.948, $p = 0.01$] and AE-related withdrawals [QM (df = 3) = 8.987, $p = 0.029$]. For treatment-related AEs, this was mainly related to a significantly higher effect-size in RCTs investigating pain conditions (regression coefficient = 0.393; $p = 0.022$) and those investigating other conditions (regression coefficient = 1.263; $p = 002$) compared to cancer conditions. For withdrawals, this was related to significantly higher effect-size in RCTs investigating pain conditions (regression coefficient = 0.816; $p = 0.020$) and multiple sclerosis (regression coefficient = 0.675; $p = 0.035$)

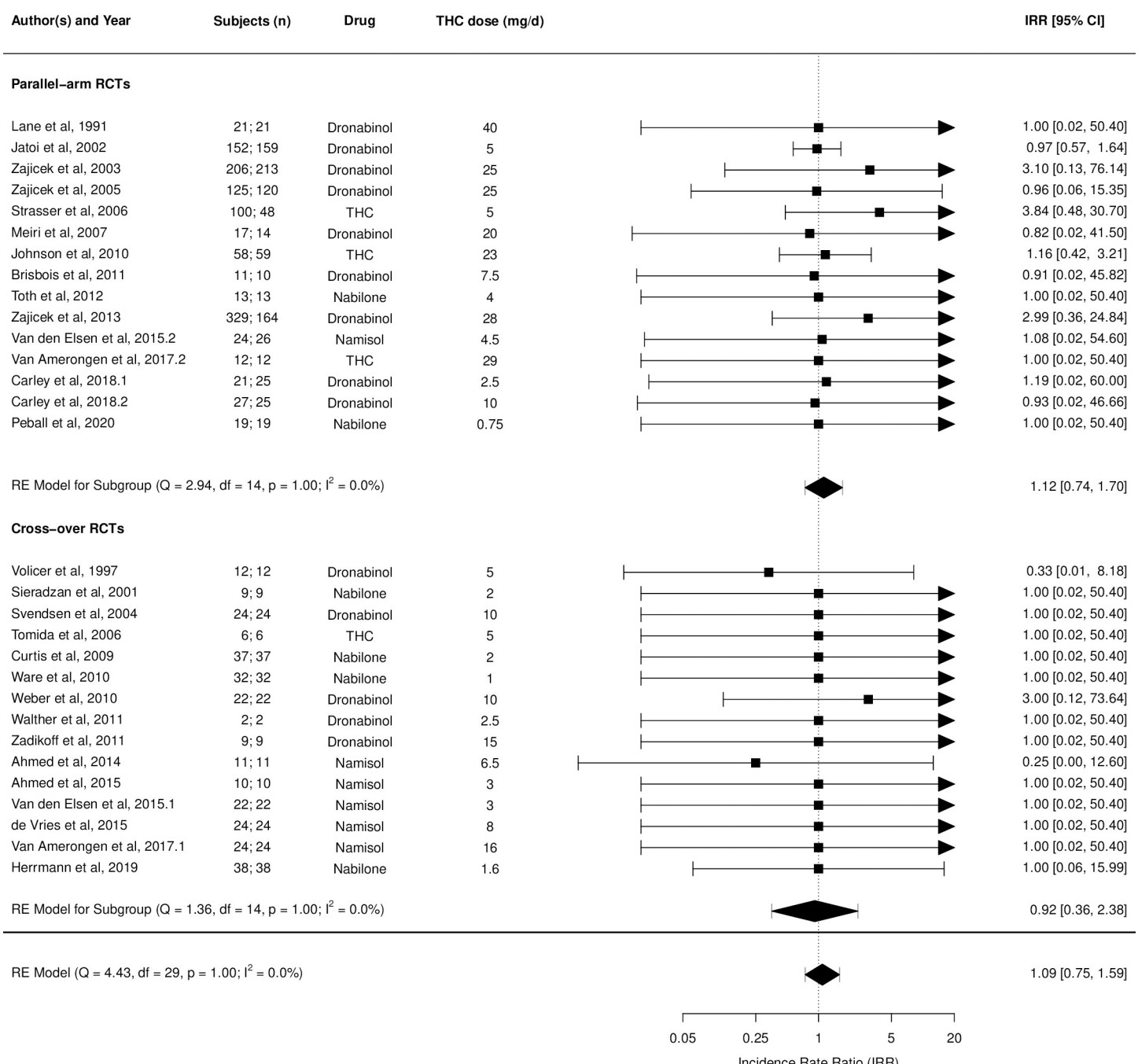

**Fig 7. Forest plot of all deaths: THC studies.** Numbers under the "Subjects (n)" column refer to analysed participants from the active and control intervention arms, respectively. IRR, incident rate ratio; RCT, randomised clinical trial; THC, delta-9-tetrahydrocannabinol.

compared to cancer conditions. Except these, moderators such as study design or type of intervention did not significantly influence estimated effects of THC:CBD combination treatment on any of the outcomes assessed.

**Effect of dose.** There was a significant effect of daily THC dose [QM (df = 1) = 4.554, $p = 0.033$] on AE-related withdrawals (Fig P in S1 Text) and a trend-level effect on all-cause

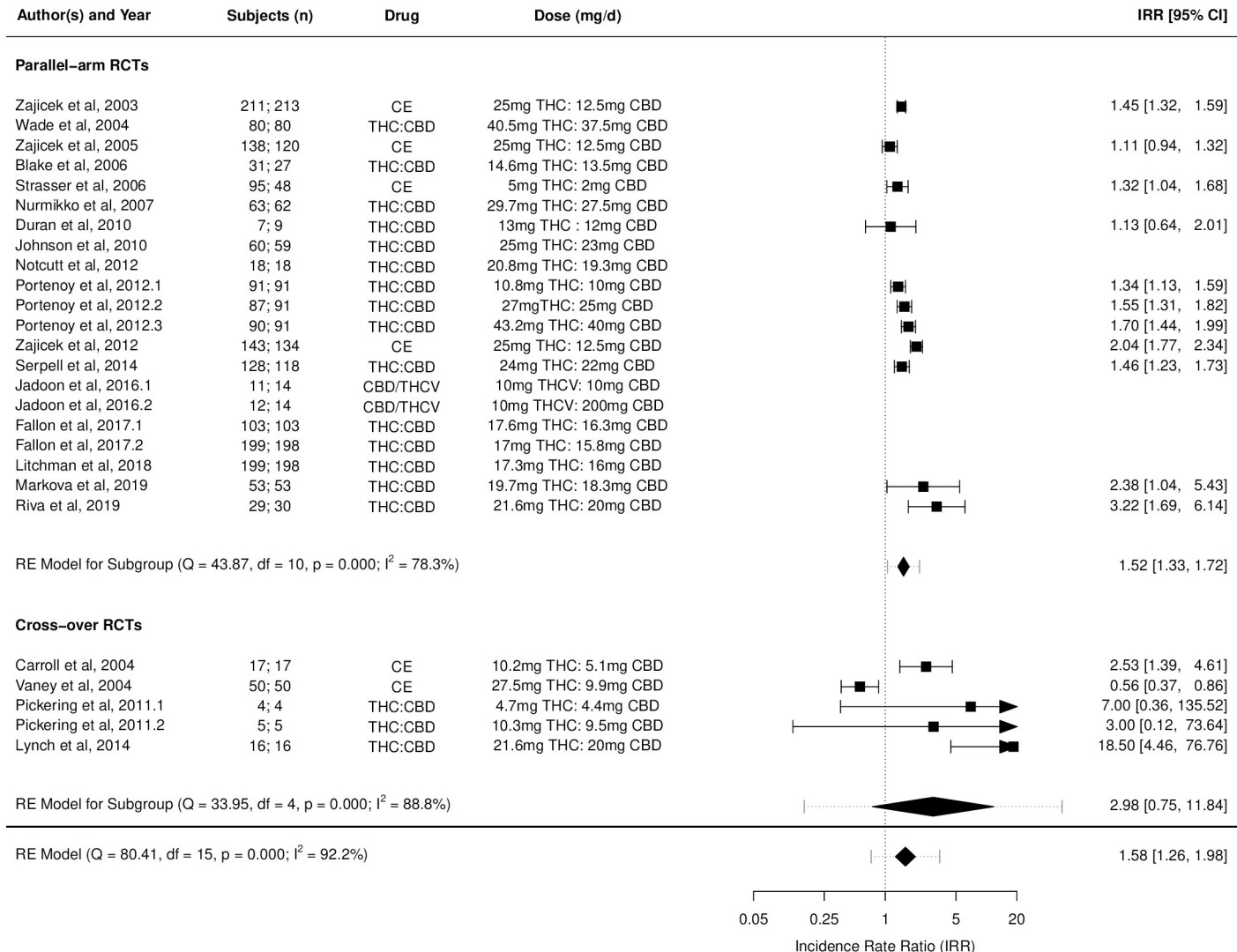

**Fig 8. Forest plot of all-cause adverse events: THC:CBD studies.** Numbers under the "Subjects (n)" column refer to analysed participants from the active and control intervention arms, respectively. CBD, cannabidiol; CE, cannabis extract; IRR, incident rate ratio; RCT, randomised clinical trial; THC, delta-9-tetrahydrocannabinol.

[QM (df = 1) = 2.899, $p$ = 0.089] and treatment-related AEs QM (df = 1) = 3.016, $p$ = 0.082]. Exploratory analyses suggested that there was also a significant effect of CBD dose (QM (df = 1) = 4.539, $p$ = 0.033) on all-cause AEs (Fig Q in S1 Text) and a trend-level effect [QM (df = 1) = 3.145, $p$ = 0.076] on treatment-related AEs but no significant effect on withdrawals. Effects of dose of both THC and CBD were such that the higher their dose, the higher was the effect of THC:CBD combination treatment on withdrawals and AEs (all-cause and treatment-related). Except these, daily THC or CBD dose did not have any significant influence on the effects of THC:CBD combination treatment on SAEs or death.

**Common side effects.** Pooled IRRs of the most commonly reported AEs (Table D in S1 Text) suggested significantly higher incidence rate of nausea, vomiting, dry mouth, dizziness/light-headedness, somnolence/drowsiness, disorientation, fatigue, and visual symptoms in active compared to control arms.

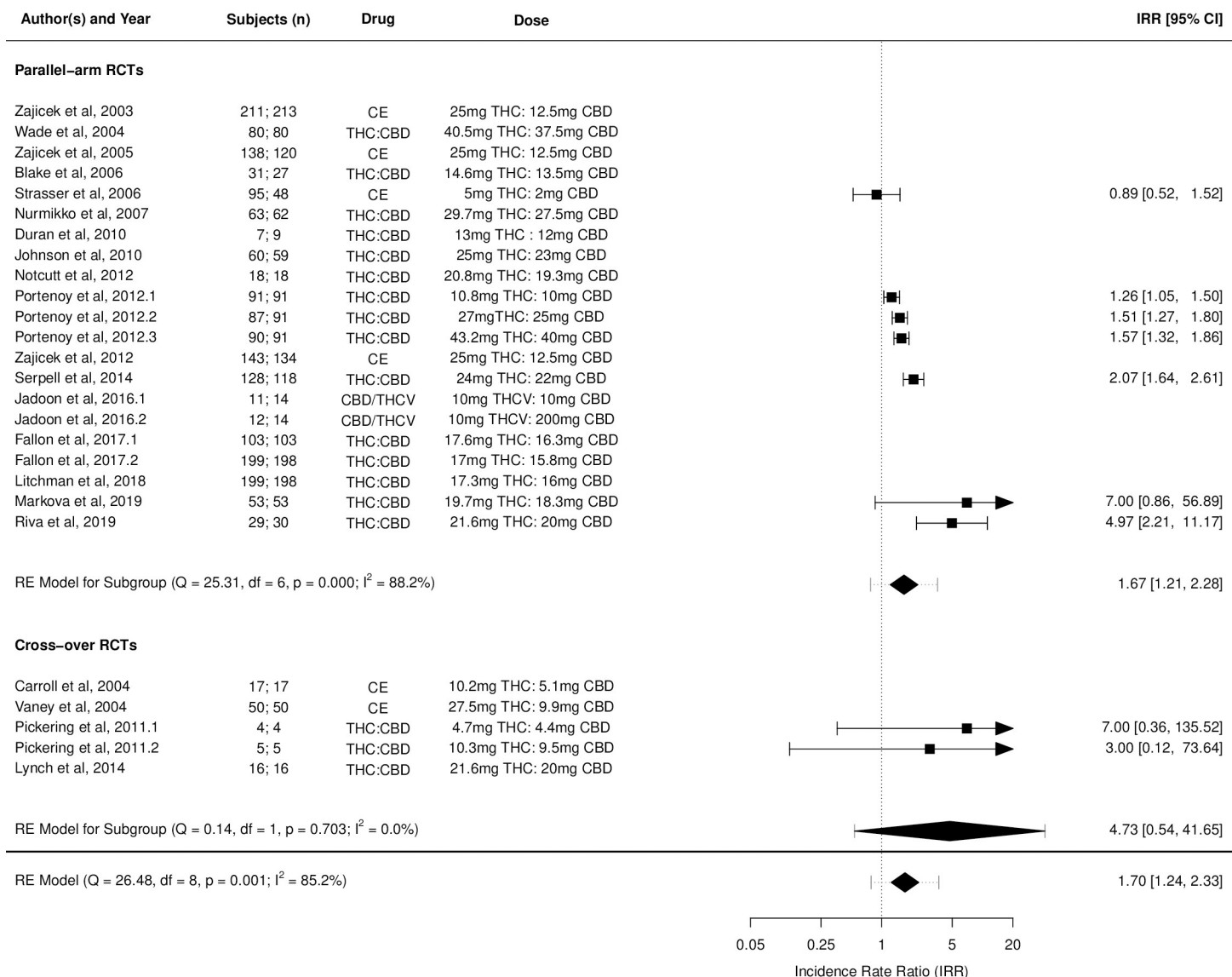

**Fig 9. Forest plot of treatment-related adverse events: THC:CBD studies.** Numbers under the "Subjects (n)" column refer to analysed participants from the active and control intervention arms, respectively. CBD, cannabidiol; CE, cannabis extract; IRR, incident rate ratio; RCT, randomised clinical trial; THC, delta-9-tetrahydrocannabinol.

**Analysis of studies where all participants were ≥50 years.** Restricting the meta-analysis to the 3 studies [42,68] that recruited participants with ≥50 years of age (total *n* = 60; analysed *n* = 52) was similar to the primary analysis in the pattern of findings, except for treatment-related AEs and AE-related withdrawals, which were no longer significant. The pooled IRRs for all-cause AEs (k = 3) and treatment-related AEs (k = 2) were 2.65 (95% CI, 1.49 to 4.71) and 4.73 (95% CI, 0.54 to 41.65), respectively. Pooled IRRs for all-cause (k = 3) and treatment-related SAEs (k = 3) were 1.00 (95% CI, 0.10 to 9.61) and 1.00 (95% CI, 0.10 to 9.61), respectively. Pooled RR for AE-related withdrawals (k = 3) and IRR for all deaths (k = 3) were 0.77 (95% CI, 0.18 to 3.22) and 1.00 (95% CI, 0.10 to 9.61), respectively (Fig R-W in S1 Text). None of these studies reported any SAEs (either treatment-related or all-cause) or death.

**Analysis of studies where all participants were ≥65 years.** As analysable data were available from only 1 study, this analysis was not carried out.

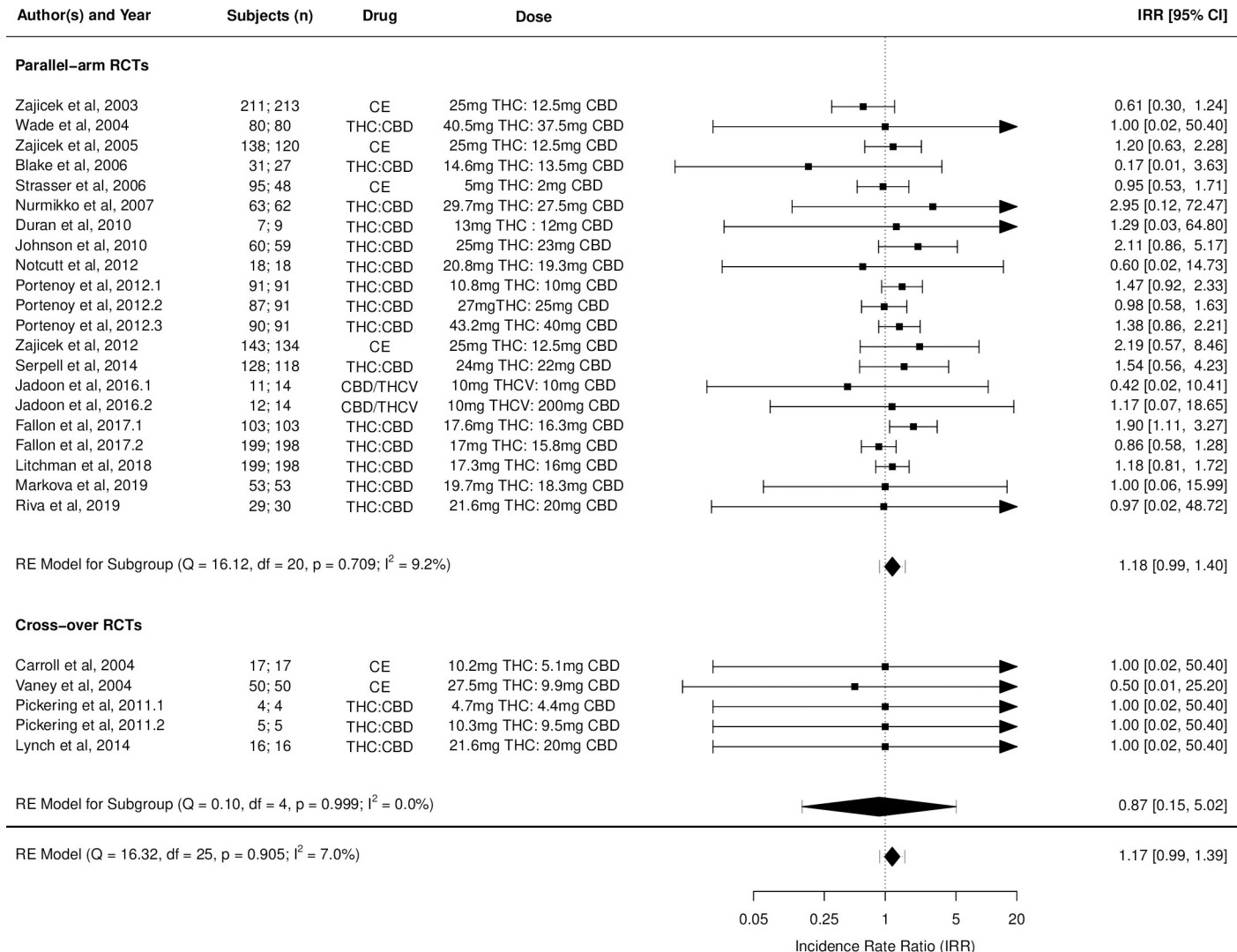

**Fig 10. Forest plot of all-cause serious adverse events: THC:CBD studies.** Numbers under the "Subjects (n)" column refer to analysed participants from the active and control intervention arms, respectively. CBD, cannabidiol; CE, cannabis extract; IRR, incident rate ratio; RCT, randomised clinical trial; THC, delta-9-tetrahydrocannabinol.

## CBD studies

Four studies (3 crossover and 1 parallel-arm RCT) from 3 articles [50,80,85] reported on 43 patients (analysed 40) on active and 44 (analysed 41) on placebo, with age ranging from 53 to 59 years [53% to 100% males; person-years of total CBD exposure: 6.60 person-years; Mean person-years of CBD exposure (mean ± SD: 1.10 ± 1.23 person-years)]. Pooled IRR for all-cause AEs for all RCTs was 1.02 (95% CI, 0.90 to 1.16), based on data available only from the 2 crossover studies reporting on 3 different dosage conditions (Fig 14; Fig X in S1 Text). There was limited data to allow quantitative synthesis of other outcomes; however, there were no treatment-related SAEs, withdrawals, or death reported (see Results in S1 Text for qualitative synthesis). No analysable data was available from studies where all study participants were ≥50 years of age.

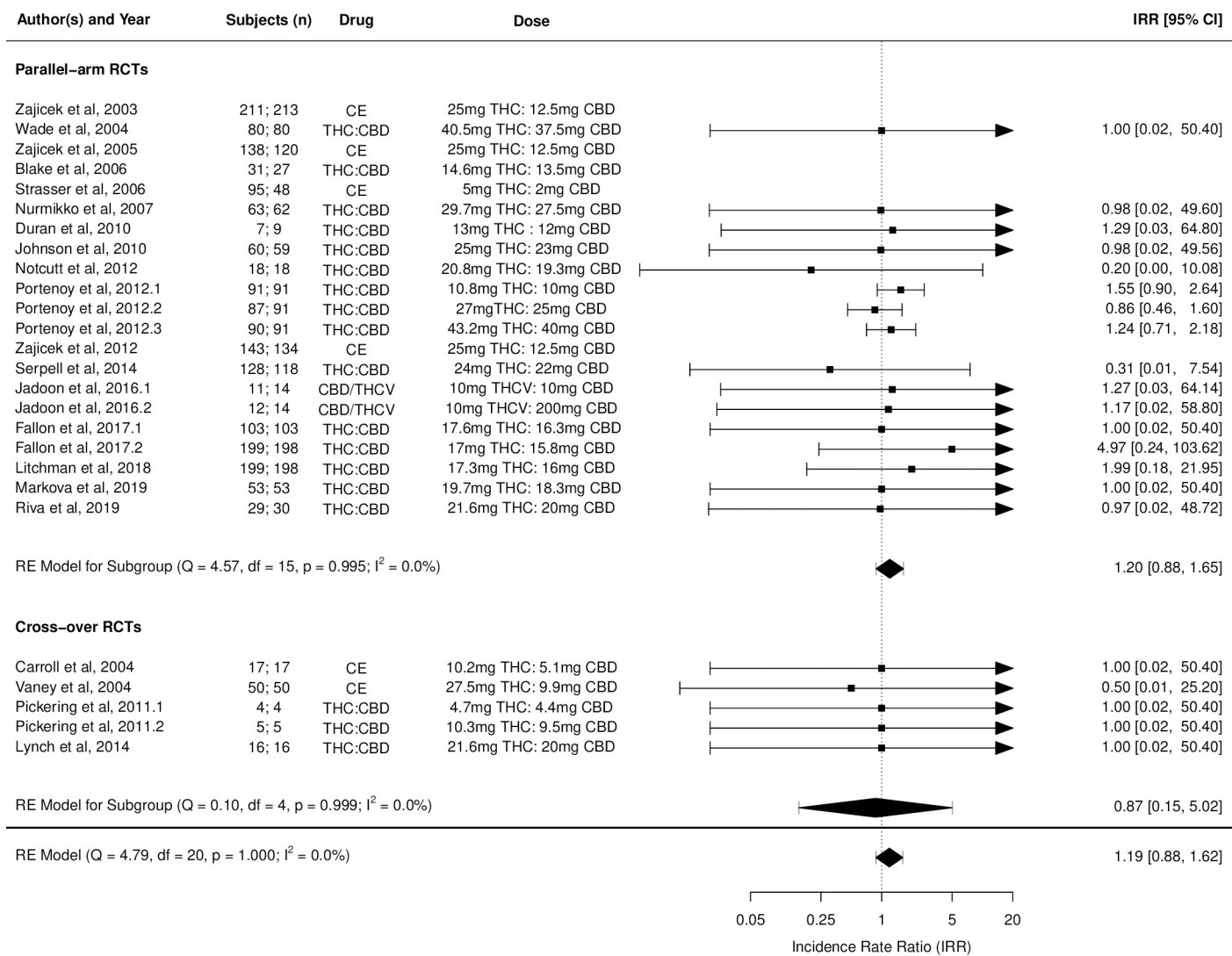

**Fig 11. Forest plot of treatment-related serious adverse events: THC:CBD studies.** Numbers under the "Subjects (n)" column refer to analysed participants from the active and control intervention arms, respectively. CBD, cannabidiol; CE, cannabis extract; IRR, incident rate ratio; RCT, randomised clinical trial; THC, delta-9-tetrahydrocannabinol.

**Risk of bias.** A summary of the risk of bias of included studies are presented in Figs 15–19. Briefly, most RCTs reported adequate randomisation sequence generation and concealment, outcome objectiveness, and masking of outcome assessors; however, some studies had high risk of bias because of potential for unmasking of participants and study personnel and selective reporting of the safety outcome. Most studies reported objective outcome assessments; however, only 60% of studies reported that outcome assessors had been appropriately blinded. A total of 45% of studies did selective reporting, i.e., they did not report data for all the safety outcomes (AEs and SAEs) in the trial and reported them when they occurred more than once or were more common or when they had occurrence above a certain threshold (1% to 10%). The authors judged 33 (55%) trials at low risk of bias, 20 (33%) trials at unclear risk of bias, and 7 (12%) trials to have high risk of bias for safety outcome reporting (Fig 15–19). Overall, 36 trials were judged to be of moderate quality, of which 15 (42%) trials reported all

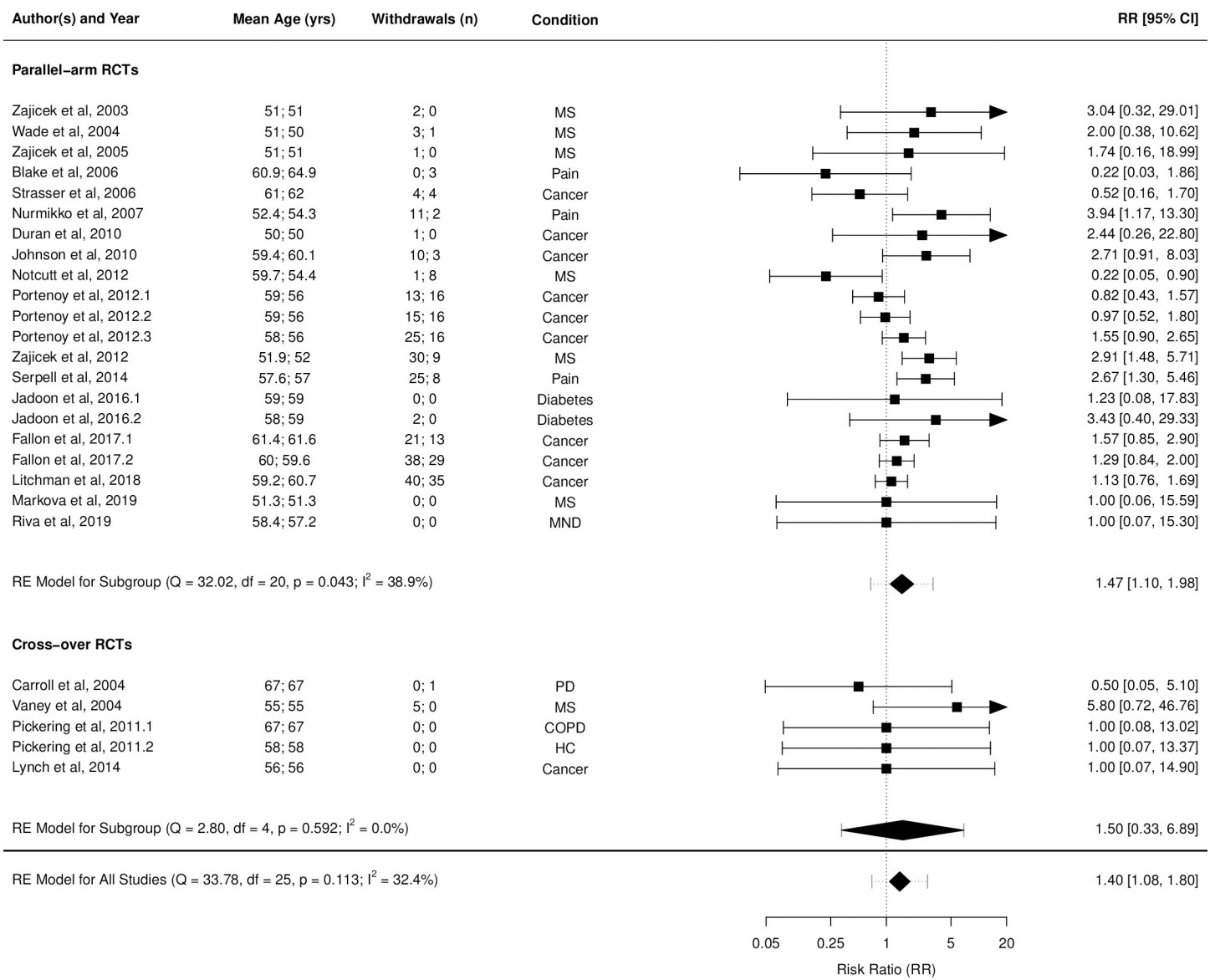

**Fig 12. Forest plot of all withdrawals: THC:CBD studies.** Numbers under the "Mean Age (yrs)" and "Withdrawals (n)" columns refer to the values in active and control intervention arms, respectively. The conditions listed are the disease conditions subgrouped for meta-regression analyses purposes are: MS; MND; pain (neuropathic pain, rheumatoid arthritis), cancer (cancer or chemotherapy-related anorexia, pain or nausea/vomiting), diabetes mellitus, COPD, HC, levodopa-induced dyskinesia in PD. CBD, cannabidiol; COPD, chronic obstructive pulmonary disease; HC, healthy controls; MS, multiple sclerosis; MND, motor neurone disease; PD, Parkinson disease; RCT, randomised clinical trial; RR, risk ratio; THC, delta-9-tetrahydrocannabinol.

AEs and SAEs. Ten (56%) of moderate-quality trials of THC only intervention reported all AEs and SAEs.

## Discussion

In this systematic review and meta-analysis, we investigated the safety and tolerability of medicinal cannabinoids in older adults by pooling data from double-blind RCTs with a reported mean participant age of 50 years and over. We hypothesized that compared to control treatments, all 3 categories of CBMs will be associated with a greater incidence of AEs, but no greater incidence of SAEs, death, or risk of withdrawal from study. We also expected a direct

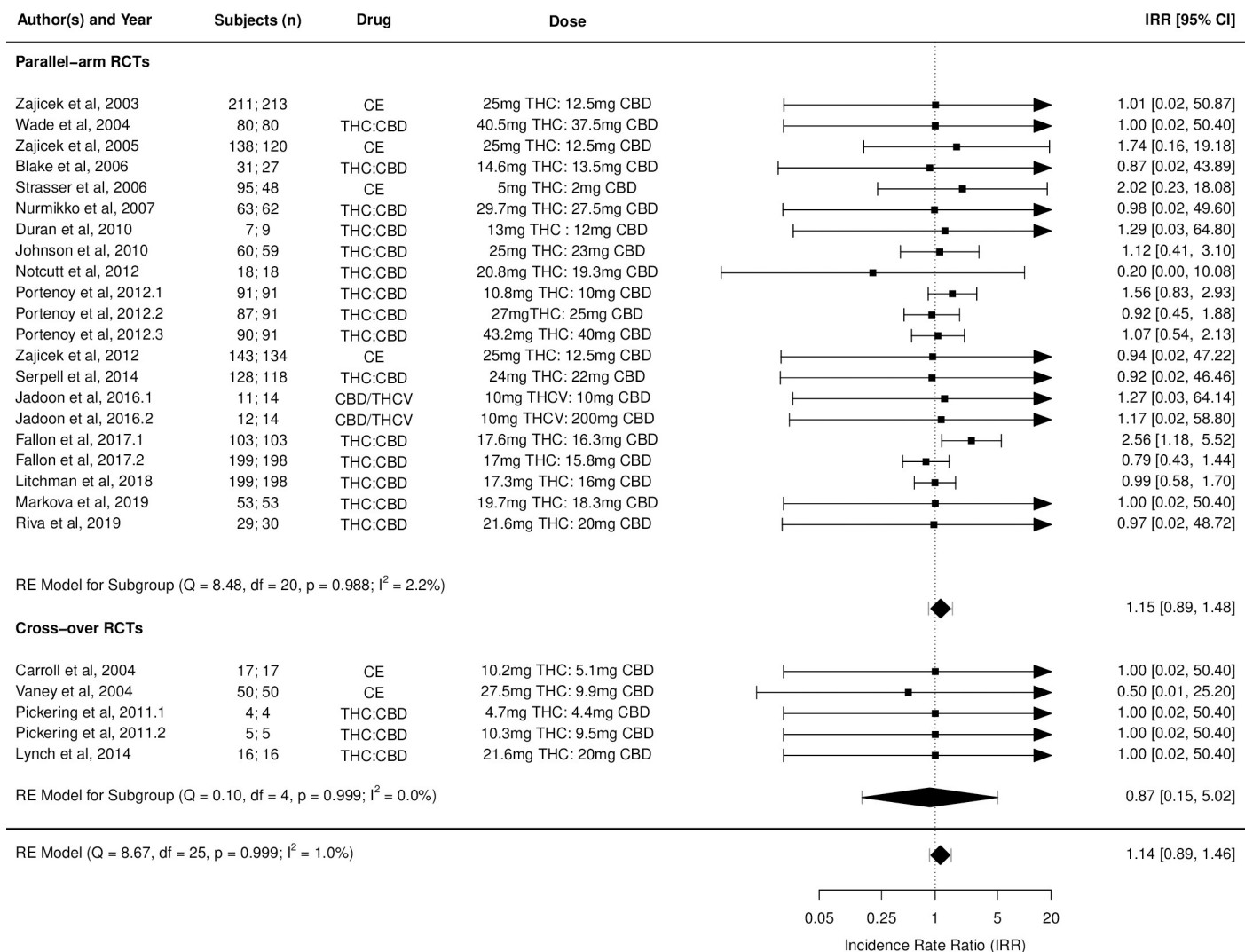

**Fig 13. Forest plot of all deaths: THC:CBD studies.** Numbers under the "Subjects (n)" column refer to analysed participants from the active and control intervention arms, respectively. CBD, cannabidiol; CE, cannabis extract; IRR, incident rate ratio; RCT, randomised clinical trial; THC, delta-9-tetrahydrocannabinol.

relationship between the dose of THC used in THC-containing CBMs and the risk of adverse outcomes. We found that generally moderate to high-quality evidence (about 60% studies) suggests that as hypothesized, treatment with THC-containing medications (THC alone and THC:CBD combination) was associated on average with significantly higher incidence of all-cause and treatment-related AEs compared to control treatments. Further, consistent with our hypotheses, the average incidence rates of serious AEs (all-cause and treatment-related) and death were not significantly greater under CBMs compared to controls in studies using THC with or without CBD. However, contrary to expectation, significantly higher risk of withdrawal related to AEs was noted on average for studies using THC:CBD combination, though this was not observed in studies that used THC without CBD. In contrast, generally low-quality evidence (about 67% of studies) suggests that CBD alone may not significantly increase the incidence rate of all cause of AEs. Qualitative synthesis of data on treatment-related AEs, SAEs, deaths, and withdrawals from study also did not suggest any increase associated with CBMs containing CBD alone. In terms of relationship between THC dose and adverse

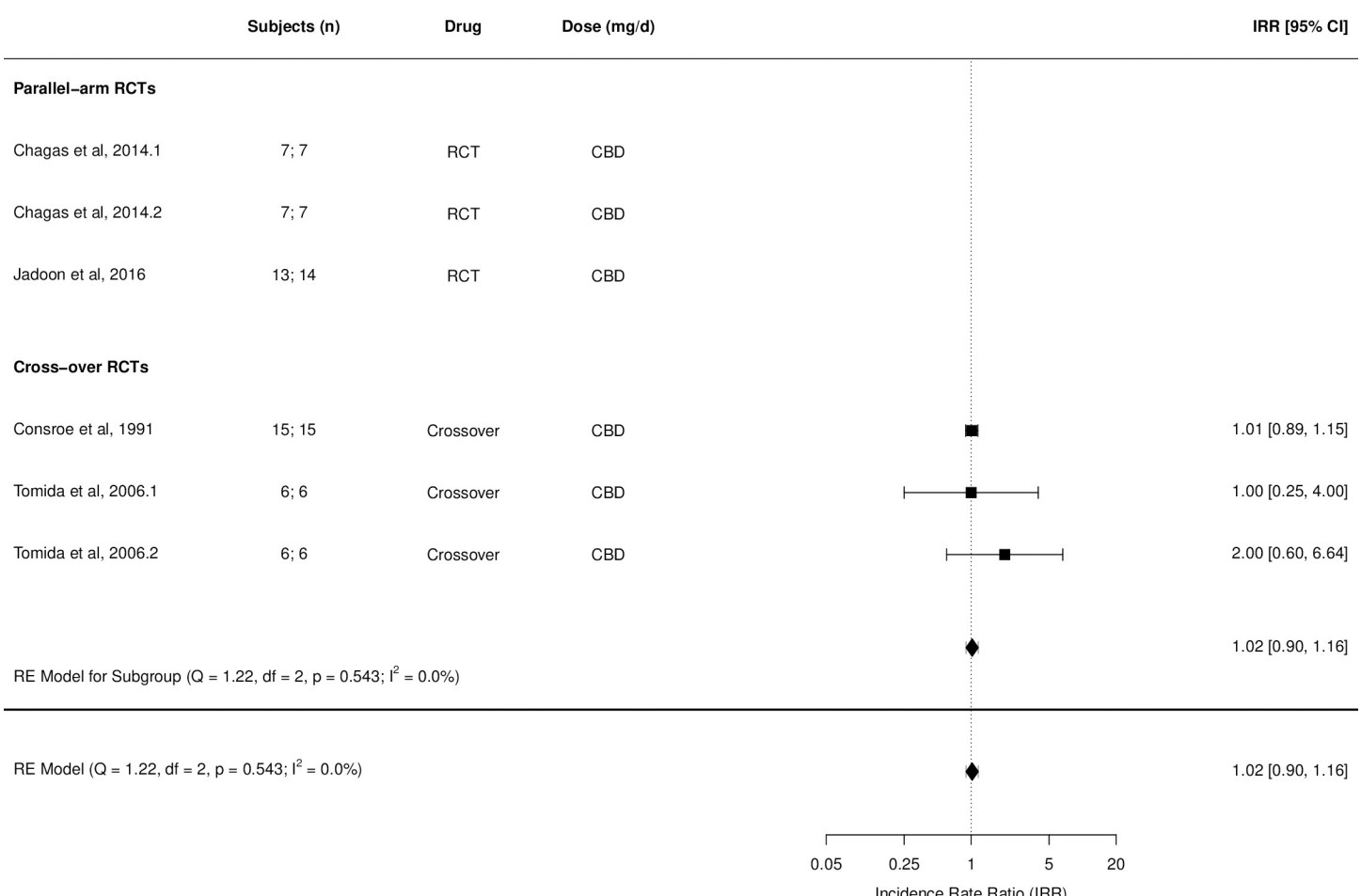

**Fig 14. Forest plot of all-cause adverse events: CBD studies.** Numbers under the "Subjects (n)" column refer to analysed participants from the active and control intervention arms, respectively. Overall study quality GRADE [31] is reported in Tables 1–3 and Results in S1 Text. Risk of bias estimates are reported in Figs 15–19. CBD, cannabidiol; GRADE, Grading of Recommendations Assessment, Development, and Evaluation; IRR, incident rate ratio; RCT, randomised clinical trial.

outcomes, as hypothesized, we found a direct relationship between daily THC dose and all-cause AEs and AE-related withdrawals in THC studies and between THC dose and AE-related withdrawals in THC:CBD studies. Contrary to our expectations, we did not find any association between THC dose and the other outcomes investigated, such as SAEs and deaths. In addition, exploratory analysis showed that CBD dose also had a significant direct relationship with all-case AEs and a strong trend-level association with treatment-related AEs in RCTs using THC-CBD combination.

Additional analyses restricted to only those studies in which all participants were ≥50 years of age or ≥65 years of age, where this was feasible, indicated a pattern of findings broadly comparable with the results of our main analyses including all studies. However, the effect of CBMs containing THC but no CBD on all-cause AEs and CBMs containing THC:CBD on treatment-related AEs and AE-related withdrawals were no longer significant, which likely reflects the lower power of these analyses which included a maximum of 4 studies in any analysis.

Collectively, our results from studies that included participants with a mean age ≥ 50 years may suggest that older adults are at significantly greater risk of both treatment-related and all-cause AEs from CBMs containing THC, but not using CBMs without THC. Despite the greater risk of AEs, older adults receiving CBMs do not seem to be at a significantly greater risk of

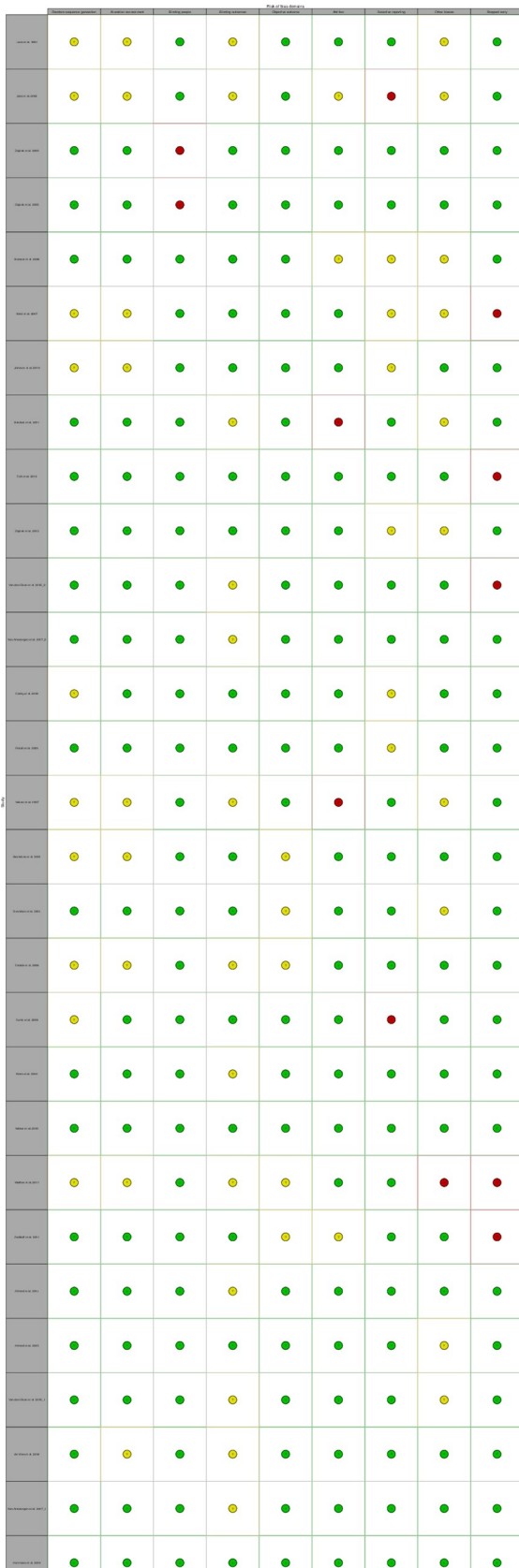

**Fig 15. Risk of bias (THC studies).** Authors' judgements about each risk of bias domain for each individual study.

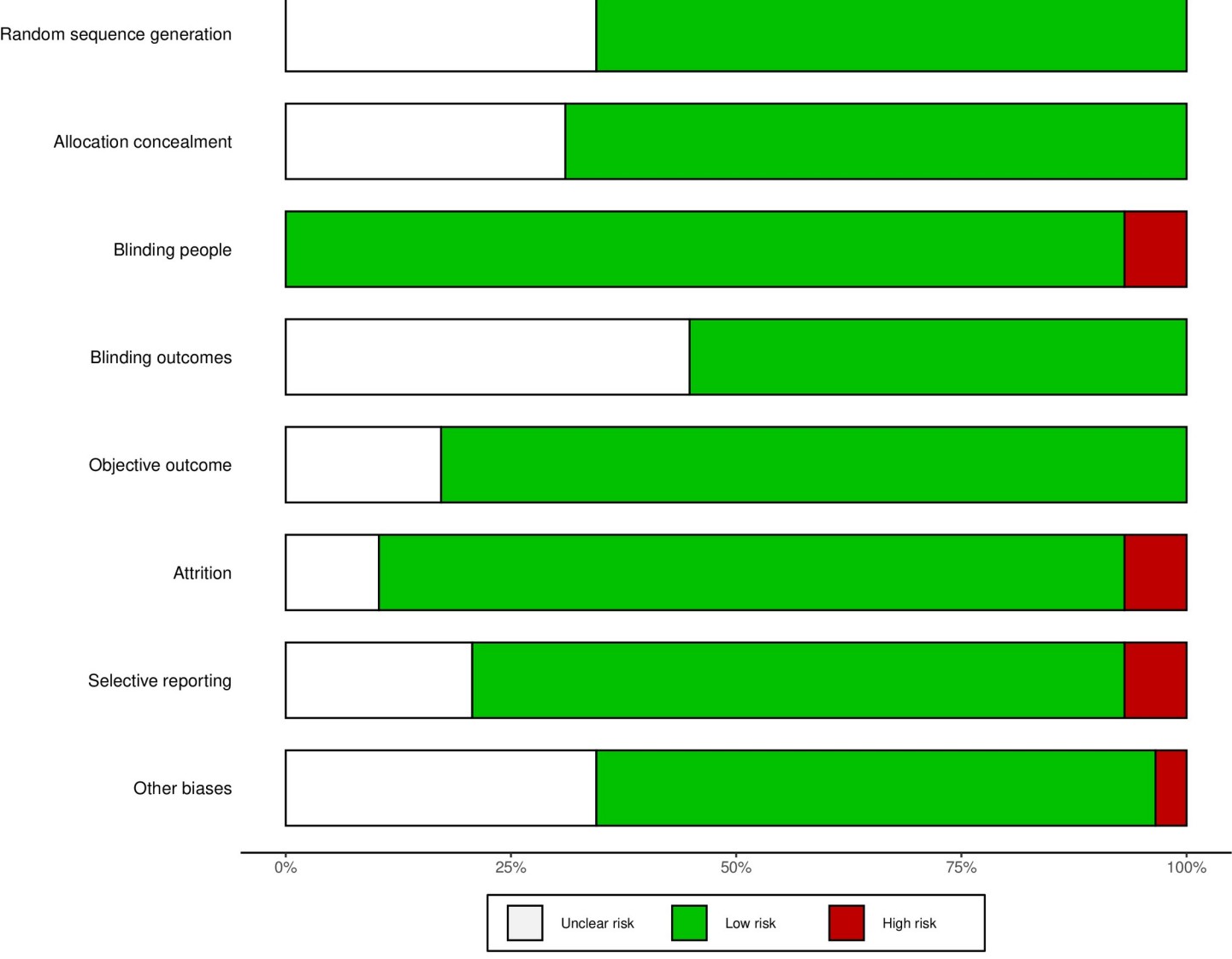

**Fig 16. Summary of risk of bias (THC studies).** Authors' judgements about each risk of bias domain reported as percentages across included studies.

more serious consequences such as SAEs or death. Greater risk of withdrawal from study in those receiving CBMs containing THC:CBD combination but not in those containing THC or CBD without the other cannabinoid is an unexpected finding. While we did not compare these 2 effects in a systematic manner, if true, this may suggest that combination of these cannabinoids may be less acceptable in this age group, who already may be receiving multiple treatments for comorbidities. What may underlie these effects is much less clear. One may speculate that this may be a result of the generally higher THC dose employed in THC:CBD studies compared to the THC studies, which may have made participants in the former group of studies more susceptible to AEs and withdrawal as a result compared to those receiving THC. This is supported by approximately double the median dose of THC used in THC:CBD studies [median and interquartile range: 10.3(10.2 to 21.6) mg/day in crossover and 20.8(14.6 to 25) mg/day in parallel-arm RCTs] as opposed to THC studies [median and interquartile range: 5(2.25 to 9 mg/day in crossover and 10(4.8 to 25) mg/day in parallel-arm RCTs]. In

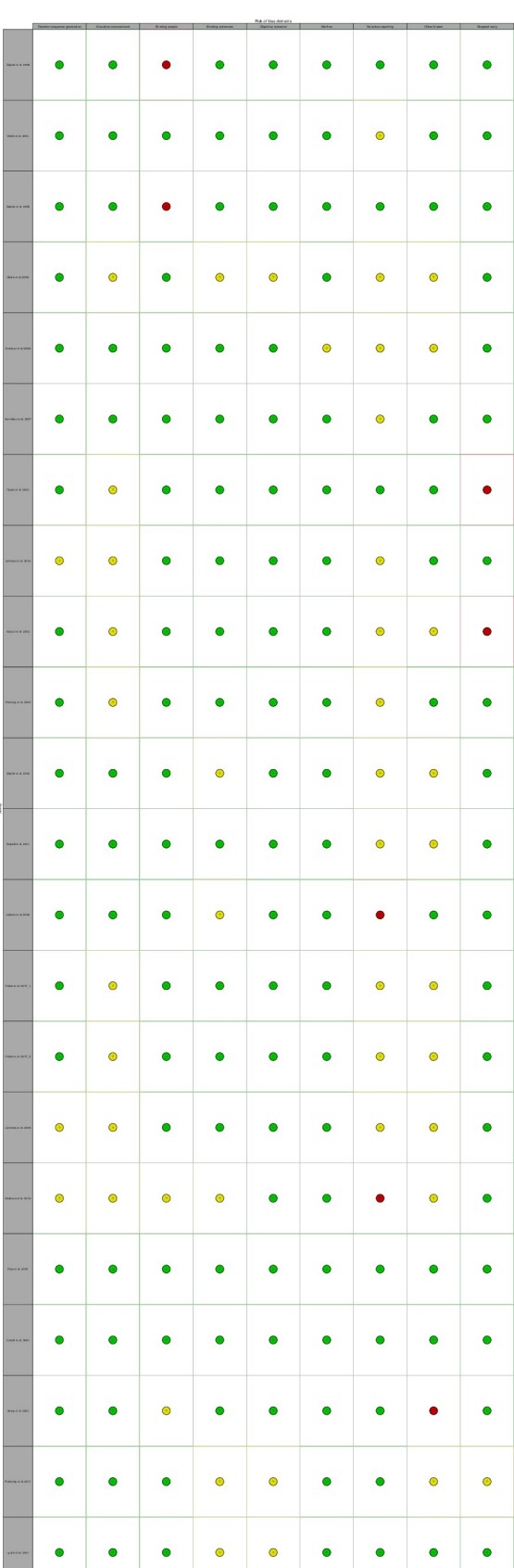

**Fig 17. Risk of bias (THC:CBD studies).** Authors' judgements about each risk of bias domain for each individual study.

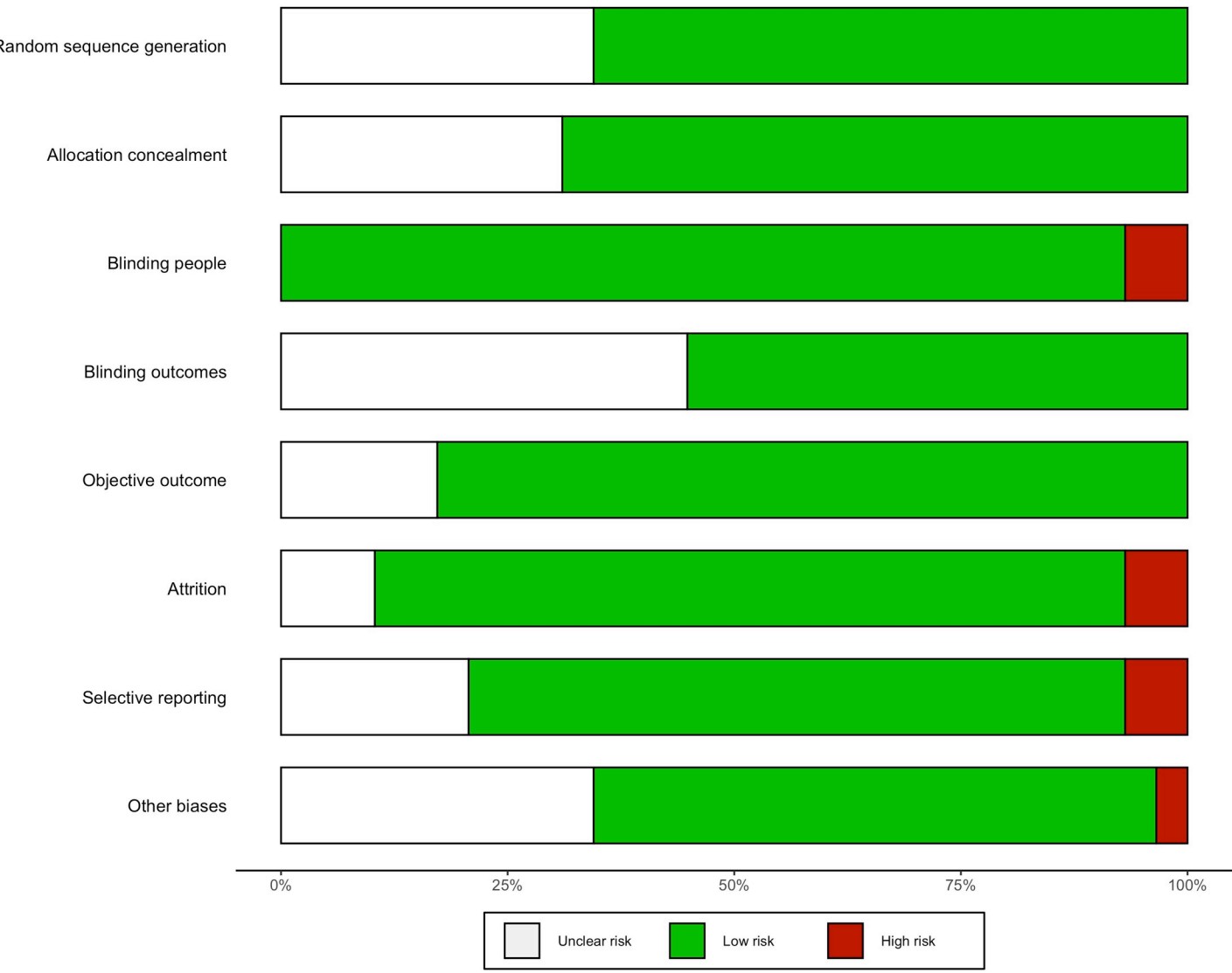

**Fig 18. Summary of risk of bias (THC:CBD studies).** Authors' judgements about each risk of bias domain reported as percentages across included studies.

accordance with this hypothesis, there was a nominally higher incidence of AEs in general in those studies using both cannabinoids in combination compared to THC studies. However, one cannot completely rule out the possibility that greater risk of withdrawal from study in those receiving CBMs containing THC:CBD combination but not in those containing THC or CBD without the other cannabinoid, as observed here, is purely by chance. Further, THC dose was significantly associated with withdrawal in both study groups using THC-containing medications. In contrast, CBD dose was only associated with AEs but not withdrawal in those using THC:CBD combination. However, whether these results suggest that lower acceptability of THC:CBD combination is a tolerability issue in older patients because of combination of cannabinoids rather than being related to the dose of CBD remains to be formally tested. It is also worth noting that the evidence base of studies that used CBMs without THC in older adults and found it to be well tolerated is relatively sparse. Whether this may underlie the absence of significantly greater risk of all-cause AEs as a result of treatment with CBMs

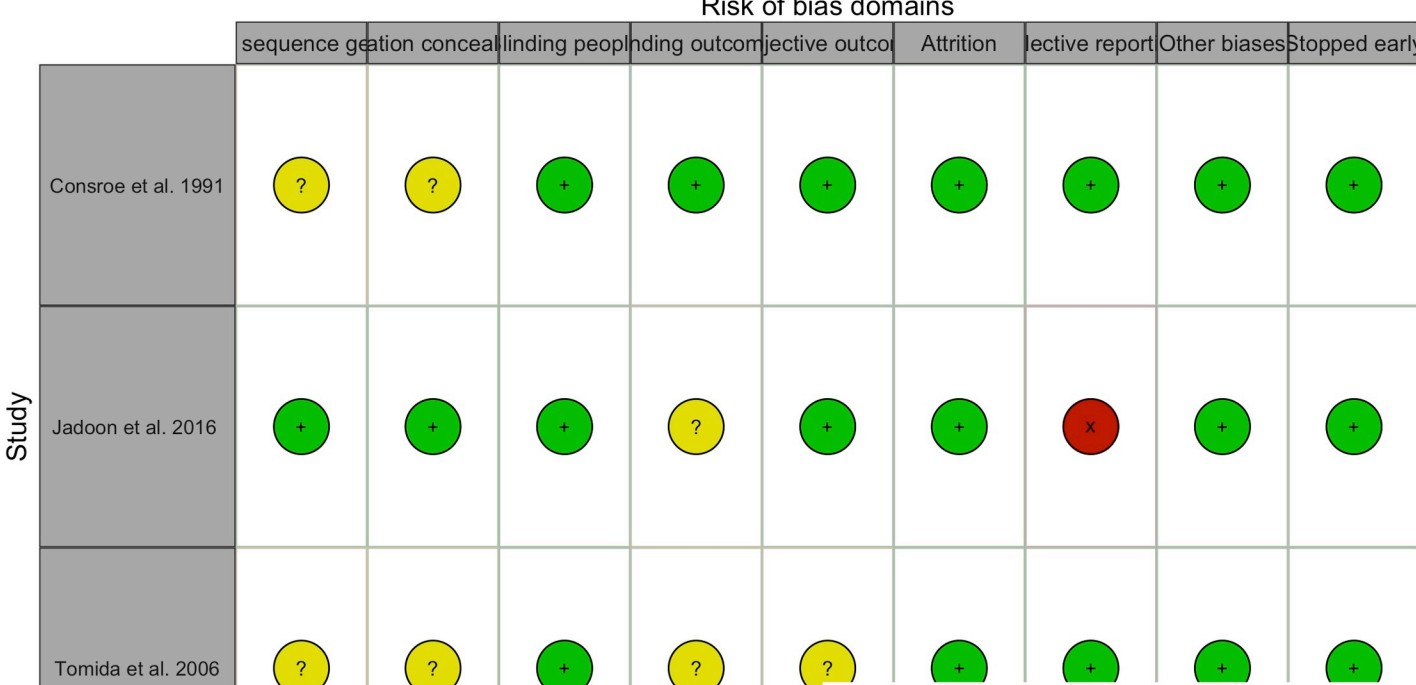

**Fig 19. Risk of bias (CBD studies).** Authors' judgements about each risk of bias domain for each individual study.

without THC compared to controls, contrary to our expectation, remains to be tested. However, more generally, lack of significant AEs associated with CBD treatment alone in the present study is consistent with other analyses suggesting a good tolerability profile of CBD alone in younger age groups [86,87]. Nevertheless, collectively, these results suggest that although the incidence of side effects is greater with THC-containing products, CBMs investigated in these studies are generally safe. Their overall tolerability profile also seems to be generally acceptable to patients, though combination of THC and CBD may be less so. It is important to note that although the general pattern of results of analyses restricted to studies where all participants were ≥50 years of age or ≥65 years of age were broadly comparable with results from our main analyses of studies with mean participant age ≥50 years, they were not identical, indicating that effects of CBMs may be different when focused exclusively on adults over 65 or 75 years of age.

Although we did observe a significant association between THC dose and all-cause AEs in THC studies, this was not present in THC:CBD studies. Whether this reflects independent evidence that CBD may mitigate some of the adverse effects of THC, such as on cognition, behaviour, autonomic, and cardiovascular function [15,25–28] and therefore may have obscured any dose–response relationship with AEs in THC:CBD studies, remains unclear. The present study was not designed to formally test this hypothesis, which needs independent examination.

Finally, we identified a number of specific AEs associated with CBMs containing THC: CBD combination or THC without CBD. Of the specific AEs, significantly increased incidence of dizziness/light-headedness, drowsiness, disorientation, and impaired mobility/balance/coordination is worth noting, in light of higher risk of falls in this age group, which may be further exacerbated by these treatments [88,89].

Pooled estimates of all-cause AEs under both THC and THC:CBD and of treatment-related AEs and AE-related withdrawals under THC:CBD as reported herein need to be considered

carefully in light of high levels of heterogeneity (discussed further below) in the studies as evident from the estimated $I^2$ statistic (>32% in all these estimates). Therefore, we have also reported the prediction intervals along with the 95% confidence intervals in the forest plots for the key pooled estimates reported herein. These prediction intervals indicate that while the average effects of CBM treatments on these outcomes are significant (as evident from the 95% confidence intervals), the range of predicted effects across different study settings that may be observed in a new study may span a wider range of effect sizes than indicated by the 95% confidence intervals. Some of these predicted effect-sizes in new studies are likely to be <1, indicating that the risk of these outcomes with THC or THC:CBD treatment may not be higher than under control treatment condition.

Previous reviews of adverse consequences of treatment with CBMs have either been qualitative [12,19,20], did not specifically focus on older adults [16,17], or did not consider the effects of THC, CBD, or their combination separately [12,16,17]. Even in studies that have pooled the adverse event data quantitatively, this has been done based on specific formulation rather than on the basis of their cannabinoid content [16,17]. Further, they have generally reported summary effect-size estimates (such as odds ratio) that do not take into account person-years of treatment, an important consideration in quantitative synthesis of outcomes from RCTs [16,17]. As in previous reviews, we found that not all studies have published all the AEs and SAEs and that there is lack of evidence of the safety, tolerability, and acceptability specifically in older patients. Nevertheless, our findings are consistent with previous reviews which found dizziness to be the most common nonserious AE with CBM treatment [12,16,17]. These results are also consistent with a prospective observational study of 901 people above 65 years of age (74.5 ± 7.5 years), who received medical cannabis from January 2015 to October 2017 in a specialized medical cannabis clinic, 31.7% reported at least 1 AE due to the treatment after 6 months, with dizziness (9.7%) and dry mouth (7.1%) as the commonest AEs [21]. Another similar study of 184 people (81.2 ± 7.5 years of age) from April 2017 to October 2018 showed 33.6% with AEs and dizziness (12.1%) and sleepiness and fatigue (11.2%) as the commonest AEs [90]. A previous systematic review of the efficacy and safety of medicinal cannabinoids which focused only on older people considered 5 controlled trials with THC ($n$ = 3) and oral THC:CBD ($n$ = 2) [12]. However, this review did not provide summary estimates (effect sizes) due to high heterogeneity among the included studies, lack of available data on means and standard deviations per treatment group, and very small sample sizes [12]. Therefore, only qualitative and descriptive summaries were provided. Studies included in this review reported dizziness, euphoria, drowsiness, confusion, and disorientation as the common adverse effects [12], consistent with our report. An earlier systematic review of AEs of medicinal cannabinoids for all ages by Wang and colleagues [17] included 23 RCTs published between 1966 and late 2007 and analysed the effect of oral THC and THC:CBD oral and spray formulations. However, they did not include data from studies examining the synthetic cannabinoid nabilone and found no evidence of higher incidence of SAEs after a median of 2 weeks use compared with a control group, regardless of the age of individuals (RR 1.04, 95% CI 0.78 to 1.30). Wang and colleagues [17] reported that respiratory, gastrointestinal, and nervous system disorders were the most frequently reported categories of SAEs, and dizziness was the most commonly reported nonserious AE (15.5%) among people exposed to cannabinoids. However, unlike the present review wherein we have used GRADE criteria to assess methodological quality of included RCTs, Wang and colleagues used a scale reported in a manuscript by Jadad and colleagues [91] to assess methodological quality of RCTs. As the scale from Jadad and colleagues does not adequately assess the quality of safety reporting in RCTs, most studies were rated as of good quality in the review by Wang and colleagues, despite their poor reporting of safety [17].

However, our results are not consistent with a more recent systematic review and meta-analysis by Whiting and colleagues [16], which included studies investigating all age groups and showed that SAEs and AE-related withdrawal are generally higher in those treated with CBMs. In contrast, we have shown that SAEs are not increased in older adults consistent with other reviews [12,17]. Whiting and colleagues [16] found pooled effect-size for any AE (OR 3.03, 95% CI 2.42 to 3.80), SAE (OR 1.41, 95% CI 1.04 to 1.92), and withdrawal due to AE (OR 2.94, 95% CI 2.18 to 3.96), and Wang and colleagues [17] found significantly higher nonserious AEs (RR 1.86, 95% CI 1.57 to 2.21) in those who received CBMs compared to control groups. In contrast to Whiting and colleagues, we have shown that increase in withdrawal may be more nuanced and more in those receiving THC:CBD combination and not for other types of CBMs.

It is difficult to compare our pooled effect-size with previous reviews due to a number of reasons. Firstly, Whiting reported odds ratios and Wang and collagues reported relative risk, wherein we have reported IRR. In the present context, the risk of AEs or SAEs in either study arm are unlikely to be constant over time. Therefore, IRR, which takes into consideration person-years of treatment and is a ratio of the incidence rate in the experimental treatment group to that in the control treatment group, is more meaningful and appropriate in contrast to the odds ratio, which is a ratio of the odds of an event in the experimental group to that in the control group and are not easily interpretable. Secondly, we carried out a pooled analysis for individual interventions unlike the other reviews, with Wang and colleagues also excluding studies using Nabilone [17]. Finally, the 2 previous reviews that reported summary estimates included all age groups and were not specific for older people unlike here.

Our results extend previous literature by showing that although at significantly higher risk of adverse events from CBMs containing THC, but not from those without THC, CBMs are generally safe in older adults, who typically experience comorbid health conditions [92] and receive polypharmacy [93]. Also, CBMs are generally acceptable as long as they are not a combination of THC and CBD.

Results presented here may also need to be considered against the side effect profile of common treatments for the clinical conditions investigated in studies included in the meta-analyses reported here, as safety and tolerability profile of a potential new treatment relative to existing alternatives is an important consideration in the context of prescribing. Studies included patients with Alzheimer disease, Parkinson disease, Huntington disease, amyotrophic lateral sclerosis, multiple sclerosis, motor neurone disease, neuropathic pain, cancer (cancer or chemotherapy-related anorexia, pain or nausea/vomiting), type 2 diabetes mellitus, chronic obstructive pulmonary disease, fibromyalgia, raised intraocular pressure, cervical dystonia, pancreatitis, and obstructive sleep apnoea. As would be expected, these conditions are typically treated with a wide range of pharmacological treatments (opioid analgesics, nonsteroidal anti-inflammatory, antiepileptics, benzodiazepines, psychotropics, cholinesterase inhibitors, glutamate antagonists, dopamine agonists, immunosuppressants, muscle relaxants, etc.) with a varied side effect profile. While a systematic comparison of side effect profile of CBMs and these other treatments would be useful and can helpfully inform prescribing decisions, this was outside the scope of the present review and was not examined here. Therefore, future studies may need to investigate this in a systematic manner. As we pooled data from randomised controlled comparisons, as opposed to observational studies without a control arm, they are very unlikely to have confounded the results presented herein.

## Strengths and limitations

Our review and meta-analysis are limited by a number of methodological weaknesses, some of which stem from the design and analytic approach of the present study and others which are

related to weaknesses in the studies that were included in the present analysis. One of the key limitations inherent in the meta-analytic approach that also applies to the present study relates to the issue of heterogeneity in the pooled data [94]. Studies pooled in our analyses focused on patients with different clinical conditions and indications, used different doses, formulations, and routes of administration of the study drug for different treatment periods and employed different study designs (crossover versus parallel-arm RCTs). Further, although we attempted to control for patients from widely disparate age groups taking part in the trial by setting an inclusion criterion of mean age of 50 years and over as the cut-off, studies included in our main analyses still involved data from people below age 50 and studies also varied in terms of sex distribution. This is reflected in the heterogeneity estimates reported in our results. In order to address this wide-ranging heterogeneity and yet pool the data in a meaningful way, we employed a random-effects model in our analyses, which assumes that there will be variability in the observed estimates of treatment effects across studies, both as a result of real differences in the effect of treatment between studies as well as by chance because of sampling variability. As a result, pooled estimates reported are not precise (as may be evident from the wide confidence intervals) and should not be considered as such. Effects reported here represent an average effect of CBM treatments on safety and tolerability outcomes investigated rather than an effect that is common across studies. As the effects may be different within an individual study, we also report a prediction interval for our key reported outcomes, in order to give an estimate of the range of predicted effects across study settings that may be observed in a new study. Further, we also carried out subgroup (e.g., crossover and parallel-arm RCTs separately) and meta-regression analyses to examine the sources of heterogeneity for THC and THC:CBD studies. These analyses did not suggest that study design or type of intervention had any significant effect on the outcomes assessed, though they suggest that effects vary, sometimes significantly, across clinical conditions investigated in the RCTs. They were more pronounced in AE-related withdrawals and treatment-related AEs in THC:CBD studies and less so for all-cause AEs in studies investigating THC as a treatment. Dose of study drug also seemed to underlie some of the heterogeneity observed in all-cause and treatment-related AEs and AE-related withdrawals in the THC and THC:CBD studies. Another limitation relates to the fact that we were not able to systematically examine the sources of heterogeneity for CBD alone studies as there were fewer studies than recommended for such analyses. One other limitation of the present meta-analysis relates to our focus on studies reporting on participants aged 50 years and over.

Further, we included studies in which the mean age of study participants was ≥50 years (although the studies also included many participants who were <50 years of age). As the cut-off employed by us differs from the conventional threshold of 65 years for "elderly" [95], this may be considered as a limitation. However, this was chosen as the clinical conditions (diabetes, cancer, neurodegenerative disorders, etc.) for which CBMs are often considered become more common from around this age. This is also a period characterised by multimorbidities, polypharmacy, and age-related bodily changes that may affect pharmacokinetics [95] and tolerability of medications. Further, in order to address the limitation that many participants in the included studies were <50 years of age, we also carried out sensitivity analysis that included studies where all participants were ≥50 years of age. The results of these analyses suggest that the pooled effect-sizes were generally in the same direction as that reported from the larger set of studies, though the confidence intervals were wider, as may be expected. We have also reported the number of studies which have actually studied individuals with age ≥65 years or ≥75 years. As evident from this, there is a very limited set of studies that have exclusively focused on people at these ages. Therefore, the present meta-analysis highlights the need for studies that may need to focus on people over 65. However, given the age range as well as

median and interquartile range of the mean ages of study participants included in the studies that constitute our meta-analysis, it is clear that people over 65 and 75 years are currently being recruited into studies of CBMs for various indications. As individual RCTs are often not powerful enough to unravel patterns of side effects and given the growing use of CBMs in the elderly and general perception of them being safe, it is particularly important to summarize currently available evidence to help inform about the safety and tolerability profile of CBMs in those aged 50 years and over rather than wait for the evidence base focusing only on ≥65 years to mature. In the fullness of time, future attempts at evidence synthesis need to focus only on studies of people ≥65 years when a sufficient number of studies have accumulated.

The other main source of limitation stems from methodological limitations in the included trials as identified during quality assessment [30,31], specifically pertaining to selective outcome reporting, and inadequate description of methods of randomization, allocation concealment, and blinding. Additionally, many included RCTs investigated only modestly sized samples [41,42,44,50,57]. Small samples render studies particularly underpowered when estimating serious and less serious adverse outcomes, which by their nature may not occur frequently. It is in this context that the present report addresses an important gap in extant evidence by systematic quantitative synthesis of RCT data following existing recommendations [30,31] to provide estimates from a larger pool of patients. Further, our analyses suggest that publication or other selection biases are unlikely to have influenced the pooled estimates reported here.

Unlike in previous meta-analyses, which reported summary effects separately based on indications, we pooled safety and tolerability data in older adults across a broad range of indications. While this may have added to the heterogeneity of the data synthesized, it allowed us to comprehensively estimate separately the effects of 3 broad categories of cannabinoid-based interventions, i.e., THC only, THC:CBD combination, and CBD only, something that has not been done before. This is a key strength of the present approach, given the reported opposite effects of different cannabinoids [4,15] that argue against data being combined. Another important strength of the present report relates to the analysis of the effects of moderators to examine the extent to which they may have influenced results, in particular relationship with cannabinoid doses used.

Studies evaluated various routes of CBM administration (oral capsules, tablets, sublingual spray, oromucosal spray). Also, not all studies compared CBMs with placebo, with 4 studies using active control treatments [43,45,54,62]. While all of these may have resulted in a very heterogeneous set of included studies, we used a random-effects model to mitigate these effects. Further, heterogeneity did not seem to significantly affect any of our estimates other than all-cause AEs, giving further confidence in the results reported. Nevertheless, we have also reported prediction intervals in addition for our key reported outcomes to give an estimate of the range of predicted effects across different study settings. Finally, another important potential limitation of the present study relates to the fact that we did not investigate the efficacy of CBMs in older adults. As outcome measures used to index efficacy vary widely between clinical conditions and there is a relative paucity of studies investigating a particular clinical condition, there are not enough data for any quantitative synthesis of efficacy of different CBMs in older adults to be meaningful just yet.

Clinical efficacy is one of the foremost considerations in addition to patient choice and safety/tolerability of interventions when prescribing in clinical practice. While the present study summarizes current evidence regarding safety/tolerability of CBMS, there is limited efficacy evidence for most clinical indications for which CBMS have been used in older people. Therefore, there is a pressing need for efficacy studies in specific indications where there is proof of concept or rationale for use of CBMs in older people. With regard to CBMs, potential

for drug–drug interaction in light of effect on cytochrome p450 enzymes is a major concern in the context of treating older patients [13]. However, few studies have examined this, an important likely determinant of tolerability and dose adjustment, and therefore worthy of investigation in future studies.

Complete reporting of safety/tolerability data as well as improved trial designs incorporating robust methods for allocation concealment, masking of participant and outcome assessors are further important considerations for future trials. Using well-powered samples, such studies need to focus on safety, tolerability, as well as efficacy of different categories of CBMs, in particular CBD on its own, a relatively less investigated CBM in older people.

## Conclusions

Results of the present study using data from RCTs with mean participant age ≥50 years suggest that although THC-containing CBMs are associated with side effects in those aged 50 years and over, in general, CBMs are safe and acceptable treatments in older adults, with a caveat that THC:CBD combinations may be less so at least in dose ranges used in studies thus far. However, tolerability may be different in adults over 65 or 75 years of age, and robust evidence of efficacy of different CBMs for specific indications is needed before they may be used in routine practice in older adults.

## Supporting information

**S1 PRISMA Checklist. PRISMA checklist.**
(DOC)

**S1 Text. Supporting Methods, Results, Tables and Figures.** Table A in S1 Text. Summary of randomised controlled trials of THC in older adults for studies with participants with mean age ≥50 years ($N = 30$) and studies with participants with age ≥50 years ($N = 4$, 13%; one ≥50 years; two ≥65 years; one ≥75 years). Table B in S1 Text. Summary of randomised controlled trials of THC:CBD combination in older adults for studies with participants with mean age ≥50 years ($N = 26$) and studies with participants recruited with age ≥50 years ($N = 3$, 11.5%; two ≥50 years; one ≥65 years). Table C in S1 Text. Summary estimates (incident rate ratio, IRR) from meta-analysis for the most commonly reported adverse events (AEs): IRR of participants experiencing AE with cannabinoid (THC) compared to placebo or active control condition. Table D in S1 Text. Summary estimates (incident rate ratio, IRR) from meta-analysis for the most commonly reported adverse events (AEs): IRR of participants experiencing AE with cannabinoid (THC:CBD combination) compared to placebo or active control condition. Table E in S1 Text. Characteristics of unpublished randomised trials of cannabinoids in older adults. **Fig A in S1 Text. Funnel plots for all tolerability and safety outcomes: THC studies.** (a) All-cause adverse events (AEs); (b) Treatment-related AEs; (c) All-cause serious adverse events (AAEs); (d) Treatment-related SAEs; (e) AE-related withdrawals; (f) deaths. Fig B in S1 Text. THC dose-related withdrawals in THC studies. Fig C in S1 Text. Forest plot of all-cause adverse events: THC studies (participants with ≥50 years of age). Numbers under the "Subjects (n)" column refer to analysed participants from the active and control intervention arms, respectively. Fig D in S1 Text. Forest plot of treatment-related serious adverse events: THC studies (participants with ≥50 years of age). Numbers under the "Subjects (n)" column refer to analysed participants from the active and control intervention arms, respectively. Fig E in S1 Text. Forest plot of all-cause serious adverse events: THC studies (participants with ≥50 years of age). Numbers under the "Subjects (n)" column refer to analysed participants from the active and control intervention arms, respectively. Fig F in S1 Text. Forest plot of treatment-

related serious adverse events: THC studies (participants with ≥50 years of age). Numbers under the "Subjects (n)" column refer to analysed participants from the active and control intervention arms, respectively. Fig G in S1 Text. Forest plot of adverse event-related withdrawals: THC studies (participants with ≥50 years of age). Numbers under the "Mean Age (yrs)" and "Withdrawals (n)" columns refer to the values in active and control intervention arms, respectively. The conditions listed are the disease conditions subgrouped into broader categories for meta-regression analyses purposes. They are: neurodegenerative disorders (ND) (dementia, Alzheimer disease, Parkinson disease (PD), Huntington disease, amyotrophic lateral sclerosis); multiple sclerosis (MS); Cancer (cancer or chemotherapy-related anorexia, pain or nausea/vomiting, chemosensory alterations); and Other (type 2 diabetes mellitus, fibromyalgia, raised intraocular pressure, cervical dystonia, healthy, pancreatitis, obstructive sleep apnoea). **Fig H in S1 Text. Forest plot of all deaths: THC studies (participants with ≥50 years of age).** Numbers under the "Subjects (n)" column refer to analysed participants from the active and control intervention arms, respectively. **Fig I in S1 Text. Forest plot of all-cause adverse events: THC:CBD studies (excluding THCV).** Numbers under the "Subjects (n)" column refer to analysed participants from the active and control intervention arms, respectively. CE refers to cannabis extract. Fig J in S1 Text. Forest plot of treatment-related adverse events: THC:CBD studies (excluding THCV). Numbers under the "Subjects (n)" column refer to analysed participants from the active and control intervention arms, respectively. CE refers to cannabis extract. Fig K in S1 Text. Forest plot of all-cause serious adverse events: THC:CBD studies. (excluding THCV). Numbers under the "Subjects (n)" column refer to analysed participants from the active and control intervention arms, respectively. CE refers to cannabis extract. Fig L in S1 Text. Forest plot of treatment-related serious adverse events: THC:CBD studies (excluding THCV). Numbers under the "Subjects (n)" column refer to analysed participants from the active and control intervention arms, respectively. CE refers to cannabis extract. **Fig M in S1 Text. Forest plot of all withdrawals: THC:CBD studies (excluding THCV).** Numbers under the "Mean Age (yrs)" and "Withdrawals (n)" columns refer to the values in active and control intervention arms, respectively. The conditions listed are the disease conditions subgrouped for meta-regression analyses purposes are: multiple sclerosis (MS); motor neurone disease (MND); pain (neuropathic pain, rheumatoid arthritis), cancer (cancer or chemotherapy-related anorexia, pain or nausea/vomiting), diabetes mellitus, chronic obstructive pulmonary disease (COPD), healthy controls (HC), levodopa-induced dyskinesia in Parkinson disease (PD). **Fig N in S1 Text. Forest plot of all deaths: THC:CBD studies (excluding THCV).** Numbers under the "Subjects (n)" column refer to analysed participants from the active and control intervention arms, respectively. CE refers to cannabis extract. **Fig O in S1 Text. Funnel plots for all tolerability and safety outcomes: THC:CBD studies.** (a) All-cause adverse events (AEs); (b) Treatment-related AEs; (c) All-cause serious adverse events (AAEs); (d) Treatment-related SAEs; (e) AE-related withdrawals; (f) deaths. Fig P in S1 Text. THC dose-related withdrawals in THC:CBD studies. Fig Q in S1 Text. CBD dose-related all-cause AEs in THC:CBD studies. Fig R in S1 Text. Forest plot of all-cause adverse events: THC:CBD studies (participants with ≥50 years of age). Numbers under the "Subjects (n)" column refer to analysed participants from the active and control intervention arms, respectively. CE refers to cannabis extract. Fig S in S1 Text. Forest plot of treatment-related adverse events: THC:CBD studies (participants with ≥50 years of age). Numbers under the "Subjects (n)" column refer to analysed participants from the active and control intervention arms, respectively. CE refers to cannabis extract. Fig T in S1 Text. Forest plot of all-cause serious adverse events: THC:CBD studies (participants with ≥50 years of age). Numbers under the "Subjects (n)" column refer to analysed participants from the active and control intervention arms, respectively. CE refers to cannabis extract. Fig U in S1 Text. Forest plot of

treatment-related serious adverse events: THC:CBD studies (participants with ≥50 years of age). Numbers under the "Subjects (n)" column refer to analysed participants from the active and control intervention arms, respectively. CE refers to cannabis extract. Fig V in S1 Text. Forest plot of all withdrawals: THC:CBD studies (participants with ≥50 years of age). Numbers under the "Mean Age (yrs)" and "Withdrawals (n)" columns refer to the values in active and control intervention arms, respectively. The conditions listed are the disease conditions subgrouped for meta-regression analyses purposes are: multiple sclerosis (MS); motor neurone disease (MND); pain (neuropathic pain, rheumatoid arthritis), cancer (cancer or chemotherapy-related anorexia, pain or nausea/vomiting), diabetes mellitus, chronic obstructive pulmonary disease (COPD), healthy controls (HC), levodopa-induced dyskinesia in Parkinson disease (PD). Fig W in S1 Text. Forest plot of all deaths: THC:CBD studies (participants with ≥50 years of age). Numbers under the "Subjects (n)" column refer to analysed participants from the active and control intervention arms, respectively. CE refers to cannabis extract. Fig X in S1 Text. Funnel plot for all-cause adverse events (AEs): CBD studies.
(DOCX)

## Acknowledgments

None.

The views expressed are those of the authors and not necessarily those of the NHS, the NIHR, PUK or the Department of Health and Social Care.

## Author Contributions

**Conceptualization:** Latha Velayudhan, Sagnik Bhattacharyya.

**Data curation:** Latha Velayudhan, Katie McGoohan.

**Formal analysis:** Sagnik Bhattacharyya.

**Methodology:** Latha Velayudhan, Katie McGoohan, Sagnik Bhattacharyya.

**Software:** Sagnik Bhattacharyya.

**Supervision:** Sagnik Bhattacharyya.

**Validation:** Latha Velayudhan.

**Writing – original draft:** Latha Velayudhan.

**Writing – review & editing:** Katie McGoohan, Sagnik Bhattacharyya.

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
