## [Editor Report · Decision Letter 0]

9 Oct 2020

Dear Dr Bhattacharyya, 

Thank you for submitting your manuscript entitled "Safety and tolerability of natural and synthetic cannabinoids in older adults: a systematic review and meta-analysis" for consideration by PLOS Medicine.

Your manuscript has now been evaluated by the PLOS Medicine editorial staff and I am writing to let you know that we would like to send your submission out for external assessment.

Kind regards,

Richard Turner, PhD

Senior editor, PLOS Medicine

rturner@plos.org

---

## [Decision Letter · Decision Letter 1]

30 Oct 2020

Dear Dr. Bhattacharyya,

Thank you very much for submitting your manuscript "Safety and tolerability of natural and synthetic cannabinoids in older adults: a systematic review and meta-analysis" (PMEDICINE-D-20-04864R1) for consideration at PLOS Medicine. 

Your paper was discussed with an academic editor with relevant expertise and sent to independent reviewers, including a statistical reviewer. The reviews are appended at the bottom of this email and any accompanying reviewer attachments can be seen via the link below:

[LINK]

In light of these reviews, we will not be able to accept the manuscript for publication in the journal in its current form, but we would like to invite you to submit a revised version that addresses the reviewers' and editors' comments fully. You will appreciate that we cannot make a decision about publication until we have seen the revised manuscript and your response, and we expect to seek re-review by one or more of the reviewers. 

We hope to receive your revised manuscript by Nov 20 2020 11:59PM. Please email us (plosmedicine@plos.org) if you have any questions or concerns.

Please let me know if you have any questions. Otherwise, we look forward to receiving your revised manuscript in due course. 

Sincerely,

Richard Turner, PhD

rturner@plos.org

Please update your search to a date in the past 3 months, say. 

Please convert your abstract into the three-part PLOS Medicine style. The final sentence of the "methods and findings" subsection should quote 2-3 of the study's main limitations. 

Are you able to quote aggregate participant demographic details in the abstract? Please add brief information about where the trials were done. 

Please begin the final subsection with "In this study, we found that ..." or similar. 

After the abstract, we ask you to add a new and accessible "author summary" section in non-identical prose. You may find it helpful to consult one or two recent research papers in PLOS Medicine to get a sense of the preferred style.

Please note in your methods section that ethics approval was not required for this analysis (assuming that this is the case). 

Please restructure the paragraph on limitations in the discussion section so as to discuss limitations of the present analysis in addition to those of the studies included. 

Please remove the information on author contributions, funding and competing interests from the end of the main text. In the event of publication, this information will appear in the article meta-data via entries in the submission form. 

Throughout the text, please convert reference call-outs to the following style: "... younger people [9,10]." (i.e., the square brackets should be preceded by a space but contain no spaces). 

In your reference list, please abbreviate journal names consistently (e.g., "PLoS Med."; and "Lancet" will suffice for reference 78).

Please adapt the attached PRISMA checklist so that individual items are referred to by section (e.g., "Methods") and paragraph number rather than by line or page numbers, as the latter generally change in the event of publication. Please refer to the checklist early in the methods section of your main text ("See S1_PRISMA_Checklist" or similar). 

Comments from Academic Editor:

1. Page 4: The authors write "THC is addictive(4)...". Please use a more specific word or set of words in lieu of the word "addictive" -- are the authors referring to physiological dependence? Psychological dependence? Both? Compulsive use despite negative consequences? Something else? 

2. Page 4: Same as above with the statement "CBD is non-addictive(5)..."

3. Page 6: As with R2, I have questions about the focus of this review to studies of "older adults", defined as studies with "mean age >=50 years". This inclusion criterion is based on an individual-level variable, ie, the authors did not restrict their meta-analysis to studies in which the age cutoff for participation was age >=50 years. Thus, many of the studies included in this meta-analysis involve much younger individuals. For example, the study by Curtis et al. 2009 had a mean age of 52 years (SD, 9.5 years), thus assuming the distribution of ages was normal we would expect approximately 16% of participants were <42 years of age. The study by Lane et al. 1991 appears to include study participants with an age range of 22-64 years. There are other examples. The authors seem to be aware of this distinction, given that in the Supplementary Appendix they write (for example): "These results are also consistent with a prospective observational study of 901 people above 65 years of age (74.5 ± 7.5 years)..."-- in that sentence it is clear that the study restricted enrollment to people >65 years of age and that the mean age of study participants was also >65 years. Yet on Page 14 the authors cite their specific focus on older-age people to distinguish their meta-analysis from two previously published meta-analyses (refs #15 and #16). The JAMA paper does not AFAIK report mean ages, but in the CMAJ paper, the mean ages for most of the included studies appear to be roughly in the same range as those included in the current meta-analysis. 

To fix this, I think the authors need to do the following:

a) I think the authors should consider restricting this meta-analysis to studies where age >=50 years was one of the inclusion criteria.

If the authors decide not to do the above, and provide reasonable justification for not doing so, then I think the authors need to do the following:

b) Make it clear in the Methods and Discussion that their inclusion criterion was *mean* age >=50 years. There are places in the manuscript where this is not always clear. For example, on page 15, the authors write: "Our use of a lower cut-off age of 50 years to define the target population..." This sentence somewhat obscures the limitation here. A more precisely worded sentence would be something like "We included studies in which the mean age of study participants was >=50 years (although the studies included many participants who were <50 years of age), and this cutoff differs from the conventional threshold of 65 years..."

c) The tables should split the rows into two categories: "Trials in which enrollment was restricted to age >=50 years" and "Trials in which enrollment was not restricted to age >=50 years, but the mean age of study participants was >=50 years".

d) If adequately powered, subgroup meta-analyses for the two categories above should be conducted (and reported in the main text if substantively different from the primary analysis; or perhaps reported in an Appendix if the pattern of findings is largely similar to the primary analysis). If subgroup analyses are not possible, the Results text should at least identify the number of RCTs and patients in which enrollment was restricted to age >=50 years.

e) A column should be added to the tables showing the age cutoff for enrollment (eg., >18 years, >40 years, "none", etc)

f) As suggested by R2, the Results text should identify the number of RCTs and patients in which enrollment was restricted to age >=65 years, and/or >=75 years. (If there were no trials in which 

g) The same modifications to the tables should be done for the Appendix Table 1. For example Appendix Tables 2A and 2B should report the N (%) of studies that were restricted to age >=50 years and age >=65 years. 

4. Page 8: The authors report that the mean age of treatment-arm study participants was "mean± SD: 44.39±190.52 person-years". How did they calculate the pooled mean age here? This detail is not reported in the Methods. They report pooled mean ages later on in the manuscript as well, eg page 11, Appendix Table 2A, etc.

5. On a related note, the authors report other pooled mean estimates (eg., Appendix Table 2A, pooled mean duration of study in weeks)-- how are these estimated?

6. Figures 2A-2G: Figure 2A reports very high levels of heterogeneity (I2>73%), but Figures 2B-2F report no heterogeneity at all (I2=0%). Is this a typo? The patterning of the figures does not seem consistent with I2's of 0%.

a) Given the high levels of heterogeneity (reported in Figure 2A at least), I would suggest including prediction intervals:

https://link.springer.com/article/10.1007/s10654-012-9738-y

b) The same modifications should be done for Appendix Figure 1A & Appendix Figures 8A-8F.

c) I think the authors should also include a more detailed discussion of the heterogeneity, which is only mentioned in passing on page 15. Some would argue that such a high degree of heterogeneity would suggest _not_ pooling the data using meta-analysis:

https://jamanetwork.com/journals/jamanetworkopen/fullarticle/2758470

https://jamanetwork.com/journals/jamanetworkopen/fullarticle/2758468

7. Figures 2G & 3G: The studies are only identified with numbers, but on first glance it is not clear what these numbers are referring to. These numbers are not listed in the other tables, obscuring the studies. I don't see the studies identified until I scrutinize the footnotes to Figures 2 & 3. 

a) The row headers for Figures 2G & 3G should instead identify the study the same way they are identifed in the other figures (eg, "Notcutt et al., 2012"). If the authors are worried about horizontal space, I would suggest formatting the figure in landscape mode.

b) In addition, the column headers for Figures 2G & 3G would benefit by simply stating the domain names (eg, "random sequence generation") instead of the abbreviation (eg., "D1"). Yes, the footnote does connect "D1" to "random sequence generation" but it is an additional cognitive step for readers. If the authors are worried about horizontal space in the column headers, they could orient the text at 90 degrees or 45 degrees.

c) The same modifications should be done for Appendix Figure 1B.

8. Appendix Table 3B & Table 4: The authors report a pooled estimate for the "immune system disorder" subgroup but the k=1-- is this a typo?

Comments from the reviewers:

*** Reviewer #1:

 I confine my remarks to statistical aspects of this paper. These were generally fine. I have one request and some comments on the figures.

First, please give an effect size whereever you give a p value (e.g. you don't do this in a couple spots on p. 11).

For the forest plots, I would combine the active and control colums (e.g 21/21) and then abbreviate the drug (D M, T, etc( to save space. Then expand teh actual graph portion. I also wonder how the numbers at teh bottom (0.02, 0.14 ...) were chosen rather than something more standard like 0.01, 0.05 etc.

But, generally, very well done

Peter Flom

*** Reviewer #2: 

This is an excellent systematic review and meta-analysis on the safety and tolerability of natural and synthetic cannabinoids in older adults.

Minor comments:

Since life expectancy have increased considerably over the past and people in the age of 50 years are nowadays commonly rather fit, the reader wonders why this age was chosen for the identification of older adults? The authors briefly address this in their discussion, however, one wonders whether other factors might have determined their choice of this age (≥ 50 years), e.g. that they would not have collected enough RCTs for this meta-analysis. Thus, they could give a little more transparency whether they would have recruited enough RCTs and patients if the cut-off in age ≥ 65 years would have been chosen.

For this reason, they should clearly state the age ≥ 50 years in their final conclusions (in the abstract and end of discussion) and state that results of this meta-analysis might be different in the age group of ≥ 65 years or even ≥ 75 years.

It is somewhat mysterious that the THC:CBD combination but not THC or CBD alone significantly increased the risk of AE-related withdrawals and the authors do not give a good explanation. Might this result be out of pure chance? The authors should give some explanation for this phenomenon?

In their supplemental discussion the authors state "Firstly, Whiting reported odds ratios and Wang et al reported relative risk, wherein we have reported IRR which takes into consideration person-years of treatment and hence is more meaningful." The authors should address this more thoroughly, explain the differences and clinical consequences in the estimated risk to the reader in the discussion of the manuscript and not only in the supplemental material.

*** Reviewer #3: 

Thank you for the opportunity to review this manuscript. The authors make a sound case for the need to determine the safety and tolerability of CBMs among older populations. My read of the review process and analysis seems appropriate, however I would encourage having a statistical expert in meta-analysis techniques to also review this paper. My comments are primarily around the positioning of the review, and how the results are contextualised.

On page 5, there is reference made to growing evidence that THC and CBD may have 'opposing effects on certain organ systems' - this should be elaborated on, both in terms of outlining what this means, and what its consequences are for patients who may use different forms of CBMs. It is also noted on page 4 that 'THC is addictive' while CBD is 'non-addictive' - I think there are many who would contest this statement. THC has largely been known as the cannabinoid that psychoactive, however there is extensive literature that maintains that cannabinoids are not addictive. If this is a statement that the authors hold, there should be a great discussion and engagement with why particular strains of cannabinoids are addictive. Giving more context to both the opposing effects on certain organ systems, as well as the debate over the different natures of CBD vs THC would help to better foreground the rationale for hypothesis i.

While it RCTs are privileged in meta-analysis reviews, it would be good if the authors could also make reference to the amount of literature that is being published such as open-label trials, n-of-1 studies, as well as observational studies where CBMs and their safety are being observed. While these methods certainly don't have the same level of rigour or control, it is worth noting, particularly in terms of how these studies align (or disagree) with the results presented in this review.

On page 6, it would be useful to give a brief overview of the broader subgroups of disease conditions. Also, if possible and to provide context, it would be useful at some point in the article to outline what the common treatments (particularly pharmaceutical) are used for these conditions, and whether treatment side effects differ greatly compared to those reported in this review.

It would be useful at the beginning of the discussion to re-state the hypotheses, and more clearly align the findings of the review to each of them. As noted, there were some findings that were contrary to predictions, and while there is some speculation for the cause of this difference, more literature could be drawn on to make sense of the finding.

There are a number of sentences throughout that read as incomplete or require a re-write that could be part of the revision process.

***

[LINK]

---

## [Decision Letter · Decision Letter 2]

9 Dec 2020

Dear Dr. Bhattacharyya,

Thank you very much for re-submitting your manuscript "Safety and tolerability of natural and synthetic cannabinoids in older adults: a systematic review and meta-analysis" (PMEDICINE-D-20-04864R2) for consideration at PLOS Medicine.

I have discussed the paper with our academic editor and it was also seen again by two reviewers. I am pleased to tell you that, provided the remaining editorial and production issues are dealt with, we expect to be able to accept the paper for publication in the journal.

[LINK]

Please let me know if you have any questions. Otherwise, we look forward to receiving the revised manuscript shortly.   

Sincerely,

Richard Turner, PhD

rturner@plos.org

Requests from Editors:

Were there any departures from the study protocol? 

Please consider adapting the title from "... older adults" to "... adults aged over 50 years". 

We would suggest also specifying the relevant age range in the first subsection of your abstract. 

Please make that "data were" in your abstract.

Please adapt your abstract so that the summary of limitations is restricted to a single sentence, beginning "Study limitations include ..." or similar and falling at the end of the "Methods and findings" subsection. Please do not include mitigating arguments in this sentence. 

Please adapt the author summary so that one or two items use the active voice, e.g. "We analysed pooled data ...". Please address "that that".

Please trim and reorganize the author summary so that it consists of up to 3 points per subsection, with each point consisting of no more than 1-2 short sentences. 

Early in the Discussion section, please adapt "We predicted ..." to "We hypothesized ..." or similar. You mention "predictions" elsewhere in the same section of the ms, and in your Introduction, and we suggest rewording here too. 

Please revisit "gender", used in the Discussion section and perhaps elsewhere, substituting "sex" where appropriate. 

Noting references 30 & 67, for example, please ensure that information on competing interests and so on is removed from all relevant citations. 

Please adapt the title of fig 1 to "Study disposition" or similar. 

RevMan permitting, please avoid "p=0.000" in fig 2a. 

Please remove the draft paper included as an attachment. 

Comments from Reviewers:

*** Reviewer #1: 

The authors have addressed my concerns and I now recommend publication.

Peter Flom

*** Reviewer #2:

Excellent revision of manscript. No more comments!

***

[LINK]

---

## [Editor Report · Decision Letter 3]

15 Dec 2020

Dear Dr Bhattacharyya, 

On behalf of my colleagues and the Academic Editor, Dr Tsai, I am pleased to inform you that we have agreed to publish your manuscript "Safety and tolerability of natural and synthetic cannabinoids in adults aged over 50 years: a systematic review and meta-analysis" (PMEDICINE-D-20-04864R3) in PLOS Medicine.

PRESS

Sincerely, 

Richard Turner, PhD 

rturner@plos.org